# EDELINE: Enhancing Memory in Diffusion-based World Models via Linear-Time Sequence Modeling

**Jia-Hua Lee**[1]    **Bor-Jiun Lin**[2]    **Wei-Fang Sun**[3]    **Chun-Yi Lee**[2]

[1] Department of Computer Science, National Tsing Hua University, Hsinchu, Taiwan
[2] Department of Computer Science, National Taiwan University, Taipei, Taiwan
[3] NVIDIA AI Technology Center (NVAITC), NVIDIA Corporation, Santa Clara, CA, USA

## Abstract

World models represent a promising approach for training reinforcement learning agents with significantly improved sample efficiency. While most world model methods primarily rely on sequences of discrete latent variables to model environment dynamics, this compression often neglects critical visual details essential for reinforcement learning. Recent diffusion-based world models condition generation on a fixed context length of frames to predict the next observation, using separate recurrent neural networks to model rewards and termination signals. Although this architecture effectively enhances visual fidelity, the fixed context length approach inherently limits memory capacity. In this paper, we introduce EDELINE, a unified world model architecture that integrates state space models with diffusion models. Our approach outperforms existing baselines across visually challenging Atari 100k tasks, memory-demanding Crafter benchmark, and 3D first-person ViZDoom environments, demonstrating superior performance in all these diverse challenges. Code is available at `https://github.com/LJH-coding/EDELINE`.

## 1 Introduction

World models [1] constitute a foundational element of modern reinforcement learning (RL) by simulating environment dynamics for agent planning and reasoning. The capacity to learn environment representations [2, 3] facilitates policy optimization through imagined trajectories, which substantially enhances sample efficiency [4] relative to conventional RL approaches. This capability is especially valuable for real-world applications in robotics and autonomous systems. Recent advances have shown exceptional performance across diverse environments, including real-world tasks [5].

Existing world models fall into two principal paradigms: *latent-space models* and *generative models*. Latent-space approaches [6, 7, 2] employ recurrent neural networks (RNNs) or variants to predict future states within a compressed latent space for efficient policy optimization. This compression, however, introduces information loss that compromises generality and reconstruction quality. Generative models, particularly diffusion-based approaches [8], have transformed world modeling through high-fidelity visual predictions via noise-reversal processes. Nevertheless, prior generative models depend on fixed-length observation-action windows that truncate historical context and fail to capture extended temporal dependencies. This limitation presents a challenge especially in partially observable environments where agents must retain and reason over prolonged observation sequences for informed decisions. Moreover, the architectural segregation of reward prediction, termination signals, and observation modeling in existing frameworks can potentially lead to suboptimal representation sharing and optimization conflicts that further impair performance.

In order to mitigate long sequence dependency issues, recent state space models (SSMs) [9, 10, 11, 12, 13] provide a complementary advantage through their capacity to model long-term dependencies efficiently. With linear-time complexity and selective state updates [12], SSMs can process theoretically unbounded sequences while preserving critical historical information. This capability is particularly valuable for world modeling, where accurate trajectory prediction often necessitates retention and reasoning across extended observation-action histories. In the world modeling domain,

recent work such as R2I [14] has addressed fundamental challenges in long-term memory and credit assignment, demonstrating the importance of enhanced temporal reasoning capabilities.

Based on these considerations, we introduce EDELINE (**E**nhancing **D**iffusion-bas**E**d World Models via **LINE**ar-Time Sequence Modeling), a unified framework that integrates the advantages of diffusion models and SSMs. EDELINE advances the state-of-the-art (SOTA) through three key innovations: (1) **Memory Enhancement**: A recurrent embedding module (REM) based on Mamba SSMs that processes unbounded observation-action sequences to enable adaptive memory retention beyond fixed-context limitations, (2) **Unified Framework**: Direct conditioning of reward and termination prediction on REM hidden states that eliminates separate networks for efficient representation sharing, and (3) **Dynamic Loss Harmonization**: Adaptive weighting of observation and reward losses that addresses scale disparities in multi-task optimization. To validate EDELINE's effectiveness, we conduct extensive evaluations. It achieves $1.87\times$ human normalized scores on the sample-efficiency challenging Atari 100k [15] benchmark and surpasses all model-based methods that do not use look-ahead search. The ablation studies confirm the effectiveness of each architectural component for world modeling performance. To substantiate EDELINE's capacity to preserve long temporal information for consistent predictions, we evaluate performance on environments that require long-term memory capability: MiniGrid-Memory [16], Crafter [17], and ViZDoom [18]. Both qualitative and quantitative results show superior temporal consistency in modeling and imaginary quality across 2D and 3D environments. Furthermore, EDELINE shows significant performance improvements compared to prior diffusion-based world models. Our contributions can be summarized as follows:

- We introduce a unified architecture EDELINE that integrates a Next-Frame Predictor for future observation imaginary, a Recurrent Embedding Module for temporal sequence processing, and a Reward/Termination Predictor to address the long-term memory limitations in existing models.

- EDELINE utilizes an SSMs-based embedding module that overcomes the fixed context limitations of prior diffusion-based methods and enhances performance in memory-demanding environments.

- Our experimental insights across various benchmarks validate EDELINE's precise prediction of reward-critical elements where prior diffusion-based world models exhibit structural inaccuracies.

## 2 Related Work

### 2.1 Diffusion Models

Diffusion models have revolutionized high-resolution image generation through their noise-reversal process. Foundational works including DDPM [19] and DDIM [20] established core principles for subsequent developments. Score-based models [21, 22] enhanced sampling efficiency through gradient estimation of data distributions, while energy-based models [23] introduced robust optimization properties via probabilistic state modeling. The application of diffusion models in RL has expanded significantly. These models serve as policy networks for efficient offline learning [24, 25, 26], enable diverse strategy generation in planning tasks [27, 28], and provide novel approaches to reward modeling [29]. MetaDiffuser [30] showed effectiveness as conditional planners in offline meta-RL, while others have adopted diffusion models for trajectory modeling and synthetic experience generation [31].

### 2.2 World Models for Sample-Efficient RL

World models serve as a fundamental component in model-based RL, enabling sample-efficient and safe learning through simulated environments. SimPLe [15] established the groundwork by introducing world models to the Atari domain and proposing the Atari 100k benchmark. Dreamer [6] advanced this field through RL from latent space predictions, which DreamerV2 [7] further refined with discrete latents to mitigate compounding errors. DreamerV3 [2] achieved a significant milestone by demonstrating human-level performance across multiple domains with fixed hyperparameters.

Recent architectural innovations have expanded world model capabilities. TWM [32] and STORM [33] incorporated Transformer architectures [34] for enhanced sequence modeling, while IRIS [35] developed a discrete image token language for structured learning. R2I [14] addressed fundamental challenges in long-term memory and credit assignment. The integration of generative modeling approaches has further advanced world model capabilities. DIAMOND [8] marked a significant breakthrough by incorporating diffusion models, which achieved superior visual fidelity and state-of-the-art (SOTA) performance on the Atari 100k benchmark. This success inspired broader applications of generative approaches in interactive environments.

## 2.3 Generative Game Engines

Another line of research explores training generative world engines using pre-collected datasets instead of RL-in-the-loop learning. GameGAN [36] pioneered this direction through GAN-based environment modeling, while Genie [37] advanced these capabilities by generating complex platformer environments from image prompts. GameNGen [38] established new standards for visual fidelity and scalability through diffusion-based environment simulation. Recently, GAIA [39, 40] and Pandora [41] have further expanded this field by developing generative world models that leverage video, text, and action inputs to produce realistic scenarios. It is important to note that these generative world methodologies do not involve RL-in-the-loop training and are therefore orthogonal to our work.

## 2.4 State Space Models (SSMs) in Reinforcement Learning

SSMs have demonstrated remarkable effectiveness in modeling long-term dependencies and structured dynamics. Foundational works [42, 9, 43, 10, 11, 44, 12, 13] introduced efficient sequence modeling approaches. Recent applications in RL include structured SSMs for in-context learning [45], DecisionMamba [46]'s adaptation of Decision Transformer [47] architecture, and Drama [48]'s integration of SSMs in world model learning.

Our work advances the state-of-the-art by synergistically combining diffusion-based world models with SSMs. While previous works have explored these approaches separately, EDELINE demonstrates that their integration enables superior temporal consistency and extended imagination horizons in world model learning. This architectural innovation enhances the visual fidelity of diffusion models through the integration of SSMs' efficient temporal processing capabilities, which addresses the key limitations of existing world model approaches.

# 3 Background

In this section, we focus on the essential concepts necessary for understanding our EDELINE framework. We provide additional background material on score-based diffusion models and multi-task world model learning in Appendix C.

## 3.1 Reinforcement Learning and World Models

The problem considered in this study focuses on image-based reinforcement learning (RL), formulated as a Partially Observable Markov Decision Process (POMDP) [49] defined by tuple $(S, A, \mathcal{O}, P, R, O, \gamma)$. Our formulation specifically considers high-dimensional image observations as inputs, as described in Section 1. The state space $S$ comprises states $s_t \in S$, while the action space $A$ can be either discrete or continuous with actions $a_t \in A$. The observation space $\mathcal{O}$ contains image observations $o_t \in \mathbb{R}^{3 \times H \times W}$. A transition function $P : S \times A \times S \rightarrow [0, 1]$ characterizes the environment dynamics $p(s_{t+1}|s_t, a_t)$, while the reward function $R : S \times A \times S \rightarrow \mathbb{R}$ maps transitions to scalar rewards $r_t \in \mathbb{R}$. The observation function $O : S \times \mathcal{O} \rightarrow [0, 1]$ establishes observation probabilities $p(o_t|s_t)$. The objective centers on learning a policy $\pi$ that maximizes the expected discounted return $\mathbb{E}_\pi[\sum_{t \geq 0} \gamma^t r_t]$, with discount factor $\gamma \in [0, 1]$. Model-based Reinforcement Learning (MBRL) [50] achieves this objective by learning a world model that encapsulates the environment dynamics $p(o_{t+1}, r_t|o_{\leq t}, a_{\leq t})$. MBRL enables learning in imagination through three systematic stages: (1) collecting real environment interactions, (2) updating the world model, and (3) training the policy through world model interactions.

## 3.2 Linear-Time Sequence Modeling with Mamba

SSMs [9] provide an alternative paradigm to attention-based architectures for sequence modeling. The Mamba architecture [12] introduces a selective state space model that offers linear time complexity and efficient parallel processing, which employs variable-dependent projection matrices to implement its selective mechanism, thus overcoming the inherent limitations of computational inefficiency and quadratic complexity in conventional SSMs [42, 9, 11, 12]. The foundational mechanism of Mamba is characterized by a linear continuous-time state space formulation via first-order differential equations as follows:

$$\frac{\partial x(t)}{\partial t} = Ax(t) + B(u(t))u(t),$$
$$y(t) = C(u(t))x(t), \tag{1}$$

where $x(t)$ represents the latent state, $u(t)$ denotes the input, and $y(t)$ indicates the output. The matrix $A$ adheres to specifications from [10]. The primary innovation compared to traditional SSMs lies in $B(u(t))$ and $C(u(t))$, which function as state-dependent linear operators to enable

selective state updates based on input content. For discretization, the system employs the zero-order-hold (ZOH) rule [51] to transform the $A$ and $B$ matrices into $\tilde{A} = \exp(\Delta A)$ and $\tilde{B} = (\Delta A)^{-1}(\exp(\Delta A) - I) \cdot \Delta B$, where the step size $\Delta$ serves as a variable-dependent parameter. This transformation enables SSMs to process continuous inputs as discrete signals and converts the original Linear Time-Invariant (LTI) equation into a recurrence format.

### 3.3 Diffusion-based World Model Learning

To adapt diffusion models for world modeling, which offers superior sample quality and tractable likelihood estimation, a key requirement is modeling the conditional distribution $p(o_{t+1}|o_{\leq t}, a_{\leq t})$, where $o_t$ and $a_t$ represent observations and actions at time step $t$. The denoising process incorporates both the noised next observation and the conditioning context as input: $D_\theta(o_{t+1}^\tau, \tau, o_{\leq t}, a_{\leq t})$. While diffusion-based world models [8] have shown promise, the state-of-the-art approach DIAMOND [8] exhibits limitations although it achieves superior performance on the Atari 100k benchmark [15]. These models face two critical limitations. The first limitation stems from their constrained conditioning context, which typically considers only the most recent observations and actions. For instance, DIAMOND restricts its context to the last four observations and actions in the sequence. This constraint impairs the model's capacity to capture long-term dependencies and leads to inaccurate predictions in scenarios that require extensive historical context. The second limitation in current diffusion-based world models lies in their architectural separation of predictive tasks. For example, DIAMOND implements a separate recurrent neural network for reward and termination prediction. This separation prevents the sharing of learned representations between the diffusion model and these predictive tasks and results in reduced overall learning efficiency of the system.

## 4 Motivational Experiments

To substantiate the memory limitations of the DIAMOND model, we conducted experiments using the MiniGrid MemoryS7 and MemoryS9 environments [16]. These experiments evaluate memory consistency and temporal prediction capabilities in world models. Fig. 1 presents qualitative comparisons among the DIAMOND model, our proposed EDELINE method, and the Oracle world model. The world models were trained on MiniGrid MemoryS7, and their state predictions were evaluated. The upper half of Fig. 1 (a) shows an initial action sequence of four consecutive left turns followed by four no-op actions. DIAMOND and EDELINE then autoregressively predicted the subsequent states. The lower half of Fig. 1 (a) reveals the prediction outcomes. At timestep 9, DIAMOND generated incorrect predictions with an extra key object in the state. The model's predictions at timesteps 10 to 11 exhibited inaccuracies in the outer wall representations. These prediction errors result from the environment's partial observability and DIAMOND's four-frame context limitation. The input of four idle frames at timestep 9 led to loss of earlier state context. In contrast, EDELINE maintained accurate predictions throughout the sequence. This improved performance stems from EDELINE's architectural design, which addresses the memory constraints inherent in DIAMOND. The accurate prediction at timestep 12 by DIAMOND can be

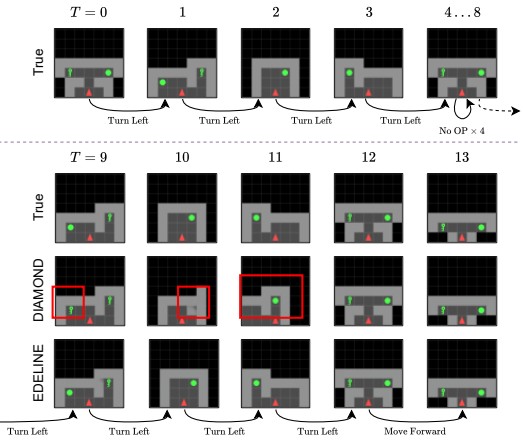

(a) Qualitative results in MemoryS7

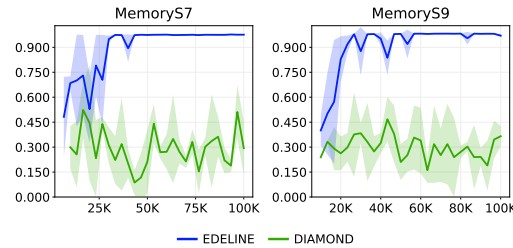

(b) Quantitative results in MemoryS7/S9

Figure 1: Motivational examples for both qualitative and quantitative evidences to demonstrate that DIAMOND faces difficulties in imagining accurate future under memorization tasks.

attributed to its access to the 8th state, which enabled accurate predictions at timestep 13 through the correct 12th state context. Fig. 1 (b) illustrates DIAMOND's performance limitations in both MemoryS7 and MemoryS9 environments due to insufficient memory capacity in partially observable scenarios. In contrast, EDELINE shows near-optimal performance under these conditions.

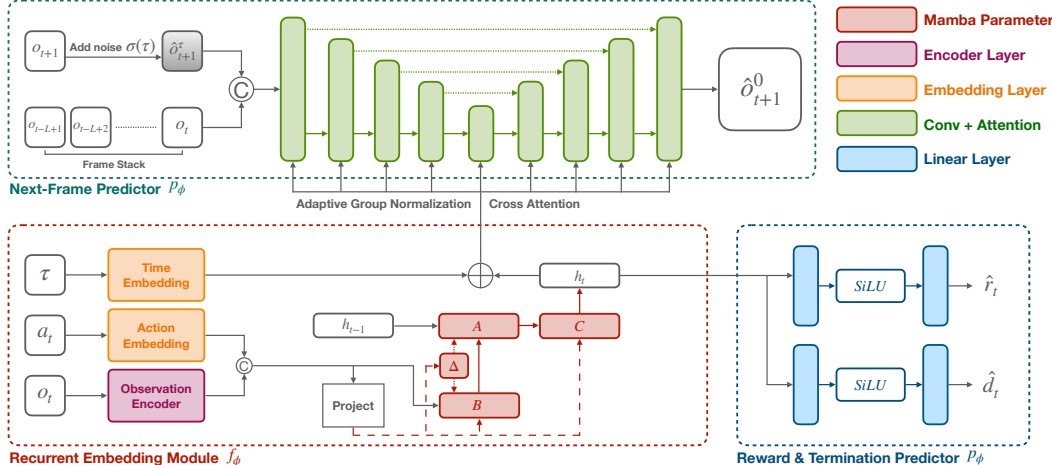

Figure 2: The EDELINE world model includes three principal components: (1) A U-Net-like *Next-Frame Predictor* enhanced by adaptive group normalization and cross-attention mechanisms, (2) A *Recurrent Embedding Module* built on Mamba architecture for temporal sequence processing, and (3) A *Reward/Termination Predictor* implemented through linear layers. The EDELINE framework uses shared hidden representations across the components for efficient world model learning.

## 5 Methodology

Conventional diffusion-based world models [8] demonstrate promise in learning environment dynamics yet face fundamental limitations in memory capacity and horizon prediction consistency. To address these challenges, this paper presents EDELINE, as illustrated in Fig. 2, a unified architecture that integrates state space models (SSMs) with diffusion-based world models. EDELINE's core innovation lies in its integration of SSMs for encoding sequential observations and actions into hidden embeddings, which a diffusion model then processes for future frame prediction. This hybrid design maintains temporal consistency while generating high-quality visual predictions. A Convolutional Neural Network based actor processes these predicted frames to determine actions, thus enabling autoregressive generation of imagined trajectories for policy optimization.

### 5.1 World Model Learning

The core architecture of EDELINE consists of a *Recurrent Embedding Module (REM)* $f_\phi$ that processes the history of observations and actions $(o_0, a_0, o_1, a_1, ..., o_t, a_t)$ to generate a hidden embedding $h_t$ through recursive computation. This embedding enables the *Next-Frame Predictor* $p_\phi$ to generate predictions of the subsequent observation $\hat{o}_{t+1}$. The architecture further incorporates dedicated *Reward and Termination Predictors* to estimate the reward $\hat{r}_t$ and episode termination signal $\hat{d}_t$ respectively. The trainable components of EDELINE's world model are formalized as:

- Recurrent Embedding Module: $h_t = f_\phi(h_{t-1}, o_t, a_t)$
- Next-Frame Predictor: $\hat{o}_{t+1} \sim p_\phi(\hat{o}_{t+1}|h_t)$
- Reward Predictor: $\hat{r}_t \sim p_\phi(\hat{r}_t|h_t)$
- Termination Predictor: $\hat{d}_t \sim p_\phi(\hat{d}_t|h_t)$

#### 5.1.1 Recurrent Embedding Module

While DIAMOND, the current state-of-the-art in diffusion-based world models, relies on a fixed context window of four previous observations and actions sequence, the proposed EDELINE architecture advances beyond this limitation through a recurrent architecture for extended temporal sequence processing. At each timestep $t$, the Recurrent Embedding Module processes the current observation-action pair $(o_t, a_t)$ to update a hidden embedding $h_t = f_\phi(h_{t-1}, o_t, a_t)$. The implementation of REM utilizes Mamba [12], an SSM architecture that offers distinct advantages for world modeling. This architectural selection is motivated by the limitations of current sequence processing methods in deep learning. Self-attention-based Transformer architectures, despite their strong modeling capabilities, suffer from quadratic computational complexity which impairs efficiency.

Traditional recurrent architectures including Long Short-Term Memory (LSTM) [52] and Gated Recurrent Unit (GRU) [53] experience gradient instability issues that affect dependency learning. In contrast, SSMs provide an effective alternative through linear-time sequence processing coupled with robust memorization capabilities via their state-space formulation. The adoption of Mamba emerges as a promising choice due to its demonstrated effectiveness in modeling temporal patterns across various sequence modeling tasks. Appendix D.8 presents a comprehensive ablation study that evaluates different architectural choices for the REM.

### 5.1.2 Next-Frame Predictor

While motivated by DIAMOND's success in diffusion-based world modeling, EDELINE introduces significant architectural innovations in its Next-Frame Predictor to enhance temporal consistency and feature integration. At time step $t$, the model conditions on both the last $L$ frames and the hidden embedding $h_t$ from the Recurrent Embedding Module to predict the next frame $\hat{o}_{t+1}$. The predictive distribution $p_\phi(o_{t+1}^0|h_t)$ is implemented through a denoising diffusion process, where $D_\phi$ functions as the denoising network. Let $y_t^\tau = (\tau, o_{t-L+1}^0, ..., o_t^0, h_t)$ represent the conditioning information, where $\tau$ represents the diffusion time. The denoising process can be formulated as $o_{t+1}^0 = D_\phi(o_{t+1}^\tau, y_t^\tau)$. To effectively integrate both visual and hidden information, $D_\phi$ employs two complementary conditioning mechanisms. First, the architecture incorporates Adaptive Group Normalization (AGN) [54] layers within each residual block to condition normalization parameters on the hidden embedding $h_t$ and diffusion time $\tau$, which establishes context-aware feature normalization [54]. This design significantly extends DIAMOND's implementation, which limits AGN conditioning to $\tau$ and action embeddings only. The second key innovation introduces cross-attention blocks inspired by Latent Diffusion Models (LDMs) [55], which utilize $h_t$ and $\tau$ as context vectors. The UNet's feature maps generate the query, while $h_t$ and $\tau$ project to keys and values. This novel attention mechanism, which is absent in DIAMOND, facilitates the fusion of spatial-temporal features with abstract dynamics encoded in $h_t$. The observation modeling loss $\mathcal{L}_{\text{obs}}(\phi)$ is defined based on Eq. (9), and can be formulated as follows:

$$\mathcal{L}_{\text{obs}}(\phi) = \mathbb{E}\left[\|D_\phi(o_{t+1}^\tau, y_t^\tau) - o_{t+1}^0\|^2\right]. \tag{2}$$

### 5.1.3 Reward / Termination Predictor

EDELINE advances beyond DIAMOND's architectural limitations through an integrated approach to reward and termination prediction. Rather than employing separate neural networks, EDELINE leverages the rich representations from its REM. The reward and termination predictors are implemented as multilayer perceptrons (MLPs) that utilize the deterministic hidden embedding $h_t$ as their conditioning input. This architectural unification enables efficient representation sharing across all predictive tasks. EDELINE processes both reward and termination signals as probability distributions conditioned on the hidden embedding: $p_\phi(\hat{r}_t|h_t)$ and $p_\phi(\hat{d}_t|h_t)$ respectively. The predictors are optimized via negative log-likelihood losses, expressed as:

$$\mathcal{L}_{\text{rew}}(\phi) = -\ln p_\phi(r_t|h_t), \mathcal{L}_{\text{end}}(\phi) = -\ln p_\phi(d_t|h_t). \tag{3}$$

This unified architectural design represents an improvement over DIAMOND's separate network approach, where reward and termination predictions require independent representation learning from the world model. The integration of these predictive tasks with shared representations enables REM to learn dynamics that encompass all relevant aspects of the environment. The architectural efficiency facilitates enhanced learning effectiveness and better performance.

### 5.1.4 EDELINE World Model Training

To implement the loss functions defined in Eq. (2) and Eq. (3), we adopt a training strategy tailored to the architecture of the EDELINE world model. The Next-Frame Predictor is trained to reconstruct future observations from hidden embeddings sampled from arbitrary timesteps. In contrast to DIAMOND, which operates over a fixed four-frame window without memory, EDELINE maintains memory across the full trajectory. Directly applying the observation loss at every timestep, as done in DIAMOND, would incur significantly higher computational costs due to EDELINE's recurrent embedding module. To mitigate this, we leverage the Mamba parallel scan algorithm to efficiently compute hidden embeddings across all timesteps, and approximate the full loss by randomly sampling a single target timestep for reconstruction. Specifically, we sample a trajectory segment of fixed

length $T$ from the replay buffer, indexed by $\mathcal{I} = \{t, \ldots, t + T - 1\}$. Given $(o_i^0, a_i)_{i \in \mathcal{I}}$, we initialize the hidden state $h_{t-1}$ and compute the hidden embeddings $\{h_i\}_{i \in \mathcal{I}}$ in parallel using Mamba. We then randomly select a target timestep $j \in \{t + \mathcal{B}, \ldots, t + T - 1\}$, where $\mathcal{B}$ denotes the burn-in period, and compute the observation reconstruction loss as:

$$\mathcal{L}_{\text{obs}}(\phi) = \|\hat{o}_j^0 - o_j\|^2, \text{ where } j \sim \text{Uniform}\{t + \mathcal{B}, \ldots, t + T - 1\}, \text{ and } \hat{o}_j^0 \sim p(\hat{o}_j^0 \mid h_{j-1}). \quad (4)$$

This approach enables EDELINE to achieve computational efficiency comparable to DIAMOND, while retaining the advantages of long-term memory. For reward and termination prediction, EDE-LINE utilizes cross-entropy losses averaged over the sampled trajectory segment, which can be formulated as follows: $\mathcal{L}_{\text{rew}}(\phi) = \frac{1}{T} \sum_{i \in \mathcal{I}} \text{CrossEnt}(\hat{r}_i, r_i)$, $\mathcal{L}_{\text{end}}(\phi) = \frac{1}{T} \sum_{i \in \mathcal{I}} \text{CrossEnt}(\hat{d}_i, d_i)$. The detailed algorithm is presented in Appendix E.

This unified training approach, combining random sampling strategies with dynamic loss harmonization, demonstrates superior efficiency compared to DIAMOND's separate network methodology, as validated in our results presented in Section 6 and Appendix D.5. Moreover, the quantitative analysis presented in Appendix D.7 reveals substantial reductions in world model training duration.

The world model integrates an innovative end-to-end training strategy with a self-supervised approach. EDELINE extends the harmonization technique from HarmonyDream [56] through the adoption of harmonizers $w_o$ and $w_r$, which dynamically balance the observation modeling loss $\mathcal{L}_{\text{obs}}(\phi)$ and reward modeling loss $\mathcal{L}_{\text{rew}}(\phi)$. This adaptive mechanism results in the total loss function $\mathcal{L}(\phi)$:

$$\mathcal{L}(\phi) = w_0 \mathcal{L}_{\text{obs}}(\phi) + w_r \mathcal{L}_{\text{rew}}(\phi) + \mathcal{L}_{\text{end}}(\phi) + \log(w_o^{-1}) + \log(w_r^{-1}). \quad (5)$$

## 5.2 Agent Behavior Learning

To enable fair comparison and demonstrate the effectiveness of EDELINE's world model architecture, the agent architecture adopts the same optimization framework as DIAMOND. Specifically, the agent integrates policy $\pi_\theta$ and value $V_\theta$ networks with REINFORCE value baseline and Bellman error optimization using $\lambda$-returns [8]. The training framework executes a procedure with three key phases: experience collection, world model updates, and policy optimization. This method, as formalized in Algorithm 1, follows the established paradigms in model-based RL literature [15, 6, 35, 8]. To ensure reproducibility, we provide extensive details in the Appendix, with documentation of objective functions, and the hyperparameter configurations in Appendices F, D.11, respectively.

# 6 Experiments

This section presents our experimental results of EDELINE. Additionally, we include ablation studies for each component of our approach in Appendix D.8. We also validate the benefits of our unified architecture through representation analysis in Appendix D.5, and evaluate the quality of our model's imagination capabilities in Appendix D.6.

## 6.1 Experimental Setup

We evaluate EDELINE on the Atari 100k benchmark [15], which serves as the standard evaluation protocol in recent model-based RL literature for fair comparison. In addition, our experimental validation extends to ViZDoom [18], MiniGrid [16], and Crafter [17] environments to demonstrate broader applicability. To ensure statistical significance, all reported results represent averages across three independent runs. The Atari 100k benchmark [15] encompasses 26 diverse Atari games that evaluate various aspects of agent capabilities. Each agent receives a strict limitation of 100k environment interactions for learning, in contrast to conventional Atari agents that typically require 50 million steps. EDELINE's performance is evaluated against current state-of-the-art world model-based approaches, including DIAMOND [8], STORM [33], DreamerV3 [2], IRIS [35], TWM [32], and Drama [48]. For evaluating 3D scene understanding capabilities, we employ ViZDoom scenarios that demand sophisticated 3D spatial reasoning in first-person environments. This provides a crucial testing ground beyond the third-person perspective of Atari environments. Furthermore, the Crafter and MiniGrid memory scenarios evaluate memorization capabilities through tasks that require information retention across extended time horizons.

Table 1: Game scores and overall human-normalized scores on the 26 games in the Atari 100k benchmark. Results are averaged over 3 seeds, with bold numbers indicating the best performing method for each metric.

| Game | Random | Human | SimPLe | TWM | IRIS | STORM | DreamerV3 | Drama | DIAMOND | EDELINE (ours) |
|---|---|---|---|---|---|---|---|---|---|---|
| Alien | 227.8 | 7127.7 | 616.9 | 674.6 | 420.0 | **983.6** | 959.4 | 820 | 744.1 | 974.6 |
| Amidar | 5.8 | 1719.5 | 74.3 | 121.8 | 143.0 | 204.8 | 139.1 | 131 | 225.8 | **299.5** |
| Assault | 222.4 | 742.0 | 527.2 | 682.6 | 1524.4 | 801.0 | 705.6 | 539 | **1526.4** | 1225.8 |
| Asterix | 210.0 | 8503.3 | 1128.3 | 1116.6 | 853.6 | 1028.0 | 932.5 | 1632 | 3698.5 | **4224.5** |
| BankHeist | 14.2 | 753.1 | 34.2 | 466.7 | 53.1 | 641.2 | 648.7 | 137 | 19.7 | **854.0** |
| BattleZone | 2360.0 | 37187.5 | 4031.2 | 5068.0 | 13074.0 | **13540.0** | 12250.0 | 10860 | 4702.0 | 5683.3 |
| Boxing | 0.1 | 12.1 | 7.8 | 77.5 | 70.1 | 79.7 | 78.0 | 78 | 86.9 | **88.1** |
| Breakout | 1.7 | 30.5 | 16.4 | 20.0 | 83.7 | 15.9 | 31.1 | 7 | 132.5 | **250.5** |
| ChopperCommand | 811.0 | 7387.8 | 979.4 | 1697.4 | 1565.0 | 1888.0 | 410.0 | 1642 | 1369.8 | **2047.3** |
| CrazyClimber | 10780.5 | 35829.4 | 62583.6 | 71820.4 | 59324.2 | 66776.0 | 97190.0 | 52242 | 99167.8 | **101781.0** |
| DemonAttack | 152.1 | 1971.0 | 208.1 | 350.2 | **2034.4** | 164.6 | 303.3 | 201 | 288.1 | 1016.1 |
| Freeway | 0.0 | 29.6 | 16.7 | 24.3 | 31.1 | 33.5 | 0.0 | 15 | 33.3 | **33.8** |
| Frostbite | 65.2 | 4334.7 | 236.9 | **1475.6** | 259.1 | 1316.0 | 909.4 | 785 | 274.1 | 286.8 |
| Gopher | 257.6 | 2412.5 | 596.8 | 1674.8 | 2236.1 | **8239.6** | 3730.0 | 2757 | 5897.9 | 6102.3 |
| Hero | 1027.0 | 30826.4 | 2656.6 | 7254.0 | 7037.4 | 11044.3 | 11160.5 | 7946 | 5621.8 | **12780.8** |
| Jamesbond | 29.0 | 302.8 | 100.5 | 362.4 | 462.7 | 509.0 | 444.6 | 372 | 427.4 | **784.3** |
| Kangaroo | 52.0 | 3035.0 | 51.2 | 1240.0 | 838.2 | 4208.0 | 4098.3 | 1384 | **5382.2** | 5270.0 |
| Krull | 1598.0 | 2665.5 | 2204.8 | 6349.2 | 6616.4 | 8412.6 | 7781.5 | 9693 | 8610.1 | **9748.8** |
| KungFuMaster | 258.5 | 22736.3 | 14862.5 | 24554.6 | 21759.8 | 26182.0 | 21420.0 | 17236 | 18713.6 | **31448.0** |
| MsPacman | 307.3 | 6951.6 | 1480.0 | 1588.4 | 999.1 | **2673.5** | 1326.9 | 2270 | 1958.2 | 1849.3 |
| Pong | -20.7 | 14.6 | 12.8 | 18.8 | 14.6 | 11.3 | 18.4 | 15 | 20.4 | **20.5** |
| PrivateEye | 24.9 | 69571.3 | 35.0 | 86.6 | 100.0 | **7781.0** | 881.6 | 90 | 114.3 | 99.5 |
| Qbert | 163.9 | 13455.0 | 1288.8 | 3330.8 | 745.7 | 4522.5 | 3405.1 | 796 | 4499.3 | **6776.2** |
| RoadRunner | 11.5 | 7845.0 | 5640.6 | 9109.0 | 9614.6 | 17564.0 | 15565.0 | 14020 | 20673.2 | **32020.0** |
| Seaquest | 68.4 | 42054.7 | 683.5 | 774.4 | 661.3 | 525.2 | 618.0 | 497 | 551.2 | **2140.1** |
| UpNDown | 533.4 | 11693.2 | 3350.3 | **15981.7** | 3546.2 | 7985.0 | 7567.1 | 7387 | 3856.3 | 5650.3 |
| #Superhuman (↑) | 0 | N/A | 1 | 8 | 10 | 10 | 9 | 7 | 11 | **13** |
| Mean (↑) | 0.000 | 1.000 | 0.332 | 0.956 | 1.046 | 1.266 | 1.124 | 0.989 | 1.459 | **1.866** |
| Median (↑) | 0.000 | 1.000 | 0.134 | 0.505 | 0.289 | 0.580 | 0.485 | 0.270 | 0.373 | **0.817** |
| IQM (↑) | 0.000 | 1.000 | 0.130 | 0.459 | 0.501 | 0.636 | 0.487 | - | 0.641 | **0.940** |
| Optimality Gap (↓) | 1.000 | 0.000 | 0.729 | 0.513 | 0.512 | 0.433 | 0.510 | - | 0.480 | **0.387** |

Figure 3: Mean, Median and IQM HNS.

## 6.2 Atari 100k Experiments

Following standard evaluation paradigms for world models, we evaluate EDELINE on the Atari 100k benchmark. For performance quantification, we adopt the human-normalized score (HNS) [57], which measures agent performance relative to human and random baselines:

$$\text{HNS} = \frac{\text{agent score} - \text{random score}}{\text{human score} - \text{random score}} \tag{6}$$

Fig. 3 presents stratified bootstrap confidence intervals following [58]'s recommendations for point estimate limitations. It can be observed that EDELINE exhibits exceptional performance across this benchmark. Our approach surpasses human players in 13 games with a superhuman mean HNS of 1.87, median of 0.82, and IQM of 0.94. These metrics demonstrate superior performance compared to existing model-based reinforcement learning baselines without look-ahead search techniques. For detailed quantitative analysis, Table 1 provides comprehensive scores for all games. The superior performance of EDELINE stems from its ability to preserve the visual fidelity advantages of diffusion-based world models. Its architecture demonstrates significant improvements over DIAMOND in memory-intensive environments such as BankHeist and Hero. The enhanced performance arises from EDELINE's memorization capabilities, combined with harmonizer-enabled precise visual detail selection for reward-relevant features. Appendices D.2, D.3, and D.4 provide additional training curves, qualitative analyses, and performance metrics, respectively.

Table 2: Comparison of different methods on Crafter in terms of average return and world model parameter count.

| Method | Avg Return | #World Model Params |
|---|---|---|
| EDELINE | **11.5 ± 0.9** | **11M** |
| DreamerV3 XL | 9.2 ± 0.3 | 200M |
| Δ-IRIS | 7.7 ± 0.5 | 25M |
| DreamerV3 M | 6.2 ± 0.5 | 37M |
| IRIS | 5.5 ± 0.7 | 48M |
| DIAMOND | 2.8 ± 0.5 | **10.4M** |

## 6.3 Crafter Experiments

To further evaluate EDELINE's memory enhancement capabilities, we conducted experiments on Crafter [17], a procedurally generated survival environment that presents complex memory challenges. Crafter was specifically designed to assess 'wide and deep exploration, long-term reasoning and credit assignment, and generalization' [2].

Crafter requires substantial memory capabilities due to its demand for agents to retain information about previously collected resources, crafted items, and explored territories for optimal decision-making, which establishes it as an ideal testbed for the evaluation of our memory-enhanced architecture. Within a 1M environment step budget, EDELINE achieves superior performance compared to state-of-the-art baselines including DIAMOND [8], DreamerV3 [2], Δ-IRIS [59], and IRIS [35], despite its relatively modest parameter count of 11M. These results highlight the significant advantages resulting from the integration of Mamba's memory capabilities with diffusion's generative abilities.

Table 2 presents our experimental results with a 1M environment step budget. The results reveal that EDELINE significantly outperforms all baselines with 25% higher returns than DreamerV3 XL despite the utilization of 18× fewer parameters. Most importantly, EDELINE delivers a 4.1× improvement over DIAMOND with a comparable parameter count, which demonstrates the substantial benefits of our enhanced memory mechanism. In order to provide additional insight into EDELINE's memory advantages, we conducted a qualitative analysis of model predictions over extended imagination. This analysis, as presented in Appendix D.10, visually demonstrates EDELINE's superior ability to remember environmental features and maintain consistency when revisiting previously explored areas, a critical capability for effective planning in complex environments.

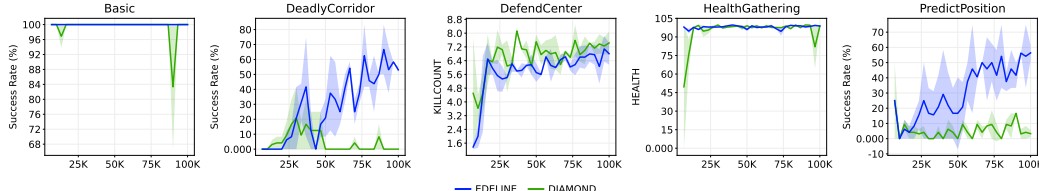

Figure 4: Training curves comparing EDELINE (blue) and DIAMOND (green) across five ViZDoom scenarios. The solid lines represent the mean performance over three seeds, with shaded regions indicating standard deviation.

## 6.4 ViZDoom Experiments

To evaluate EDELINE's memory retention and imagination consistency in more challenging scenarios, we conduct experiments in the first-person 3D environment ViZDoom, which presents increased complexity compared to Atari 100k's two-dimensional perspective. Our evaluation encompasses five default scenarios (i.e., Basic, DeadlyCorridor, DefendCenter, HealthGathering, PredictPosition), with detailed scenario descriptions and reward configurations available in Appendix D.9. The experimental results presented in Fig. 4 demonstrate EDELINE's superior performance over DIAMOND in scenarios that demand sophisticated 3D scene modeling (DeadlyCorridor) and spatiotemporal prediction (PredictPosition).

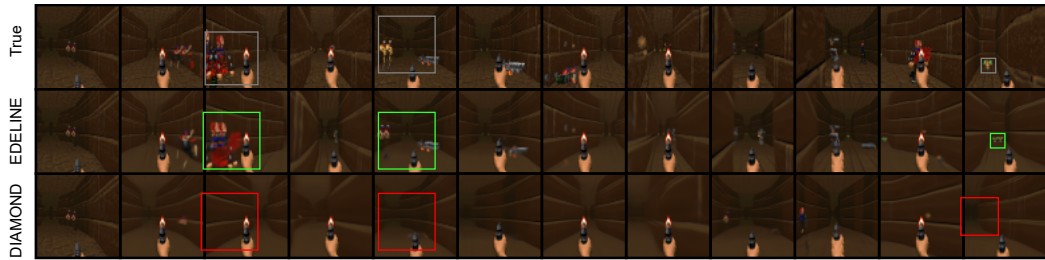

Figure 5: Qualitative comparison on the DeadlyCorridor scenario. Each row shows ground truth, EDELINE's predictions, and DIAMOND's predictions, respectively. Colored boxes highlight successful (green) and failed (red) predictions of task-critical visual elements including enemies, particle effects from hits, and the armor. EDELINE accurately predicts the reward-relevant visual details. DIAMOND captures only basic environment structure.

The DeadlyCorridor scenario provides a particularly compelling demonstration of EDELINE's memory advantages. This environment requires the agent to navigate a corridor, eliminate enemies on both sides, and ultimately reach armor at the corridor's end. The task necessitates sustained spatial awareness throughout a relatively long trajectory. As Fig. 5 illustrates, EDELINE's SSM effectively integrates information from the entire history, whereas DIAMOND remains constrained to only the past four frames. This memory limitation significantly impairs DIAMOND's ability to maintain consistent predictions in 3D environments where current observations provide only partial information about the scene. The visual comparison reveals that EDELINE accurately models both the game environment and particle effects while correctly predicts the corridor's end and armor object in the final steps. This substantiates EDELINE's comprehension of the agent's position relative to the corridor's end through its extended memory capabilities, while DIAMOND's predictions become increasingly blurry and inconsistent due to its memory constraints. It also reveals that the enhanced spatial awareness directly contributes to EDELINE's superior performance in the environment.

# 7 Conclusions

In this work, we addressed the limitations of current diffusion-based world models in handling long-term dependencies and maintaining prediction consistency. Through the integration of Mamba SSMs, EDELINE effectively processed extended observation-action sequences through its recurrent embedding module, which enabled adaptive memory retention beyond fixed-context approaches. The unified framework eliminated architectural separation between observation, reward, and termination prediction, which fostered efficient representation sharing. Dynamic loss harmonization further mitigated optimization conflicts arising from multi-task learning. Extensive experiments on Atari 100K, Crafter, MiniGrid, and ViZDoom validated EDELINE's state-of-the-art performance, with significant improvements in quantitative metrics and qualitative prediction fidelity.

## Acknowledgements

The authors gratefully acknowledge the support from the National Science and Technology Council (NSTC) in Taiwan under grant numbers NSTC 114-2221-E-002-069-MY3, NSTC 113-2221-E-002-212-MY3, and NSTC 114-2218-E-A49-026. We also express our sincere appreciation to NVIDIA Corporation and the NVIDIA AI Technology Center (NVAITC) for the donation of GPUs and access to the Taipei-1 supercomputer. Furthermore, the authors extend their gratitude to the National Center for High-Performance Computing for providing the necessary computational and storage resources.

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

# A  Limitation

Although EDELINE maintains computational efficiency equivalent to that of DIAMOND and simultaneously addresses its memory limitations, the fundamental computational requirements of diffusion-based world models remain substantial relative to alternative approaches. The iterative characteristics of the diffusion process necessitate multiple forward passes throughout both training and inference phases, which consequently elevates computational overhead. The enhancement of computational efficiency within diffusion processes for world modeling applications constitutes a promising avenue for future research endeavors.

# B  Broader Impact

EDELINE addresses fundamental limitations inherent in diffusion-based world models through the enhancement of memory capabilities and imagination consistency. Our research contributes three pivotal advances to data-efficient reinforcement learning: (1) we overcome the fixed context window constraints of prior methodologies such as DIAMOND through the implementation of selective state space models, which substantially enhances memory capabilities, (2) we establish an end-to-end training methodology that effectively leverages hidden embeddings and incorporates HarmonyDream to augment world modeling performance, and (3) we demonstrate superior performance across visually rich 2D environments (i.e., Atari), challenging 3D first-person perspectives (i.e., ViZDoom), and memory-intensive procedurally-generated survival scenarios (i.e., Crafter), which illustrates the potential of our approach to address increasingly complex domains.

World models represent a promising direction for the improvement of sample efficiency and safety in reinforcement learning through the reduction of direct environment interaction requirements. While EDELINE advances this research domain, we acknowledge that imperfections in world models may continue to result in suboptimal agent behaviors. We believe our contributions toward more accurate and memory-capable world models will serve to mitigate such risks in future applications.

# C   Additional Background Material Section

## C.1   Score-based Diffusion Generative Models

Diffusion probabilistic modeling [60, 19, 61] and score-based generative modeling [21, 62, 63] can be unified through a forward stochastic differential equation (SDE) formulation [22]. The forward diffusion process $\{\mathbf{x}^\tau\}$ with continuous time variable $\tau$ transforms the data distribution $p^0 = p^{\text{data}}$ to prior distribution $p^T = p^{\text{prior}}$, expressed as:

$$d\mathbf{x} = \mathbf{f}(\mathbf{x}, \tau)d\tau + g(\tau)d\mathbf{w}, \tag{7}$$

where $\mathbf{f}(\mathbf{x}, \tau)$ represents the drift coefficient, $g(\tau)$ denotes the diffusion coefficient, and $\mathbf{w}$ is the Wiener process. The corresponding reverse-time SDE can then be formulated as:

$$d\mathbf{x} = \left[\mathbf{f}(\mathbf{x}, \tau) - g(\tau)^2 \nabla_\mathbf{x} \log p^\tau(\mathbf{x})\right] d\tau + g(\tau)d\bar{\mathbf{w}}, \tag{8}$$

where $\bar{\mathbf{w}}$ is the reverse-time Wiener process. Eq. (8) enables sampling from $p^0$ when the (Stein) score function $\nabla_\mathbf{x} \log p^\tau(\mathbf{x})$ is available. A common approach to estimate the score function is through the introduction of a denoiser $D_\theta$, which is trained to minimize the following objective:

$$\mathbb{E}_{\sigma \sim p^{\text{train}}} \mathbb{E}_{\mathbf{x}^0 \sim p^{\text{data}}} \mathbb{E}_{\mathbf{n} \sim \mathcal{N}(0, \sigma^2 I)} \left[\|D_\theta(\mathbf{x}^0 + \mathbf{n}; \sigma) - \mathbf{x}^0\|_2^2\right], \tag{9}$$

where $\mathbf{n}$ is Gaussian noise with zero mean and variance determined by a variance scheduler $\sigma(\tau)$ that follows a noise distribution $p^{\text{train}}$, and $(\mathbf{x}^0 + \mathbf{n})$ corresponds to the perturbed data $\mathbf{x}^\tau$. The score function can then be estimated through: $\nabla_\mathbf{x} \log p^\tau(\mathbf{x}) = \frac{1}{\sigma^2}(D_\theta(\mathbf{x}; \sigma) - \mathbf{x})$. In practice, modeling the denoiser $D_\theta$ directly can be challenging due to the wide range of noise scales. To address this, EDM [64] introduces a design space that isolates key design choices, including preconditioning functions $\{c_{\text{skip}}, c_{\text{out}}, c_{\text{in}}, c_{\text{noise}}\}$ to modulate the unconditioned neural network $F_\theta$ to represent $D_\theta$, which can be formulated as:

$$D_\theta(\mathbf{x}; \sigma) = c_{\text{skip}}(\sigma)\mathbf{x} + c_{\text{out}}(\sigma)F_\theta(c_{\text{in}}(\sigma)\mathbf{x}; c_{\text{noise}}(\sigma)). \tag{10}$$

The preconditioners serve distinct purposes: $c_{\text{in}}(\sigma)$ and $c_{\text{out}}(\sigma)$ maintain unit variance for network inputs and outputs across noise levels, $c_{\text{noise}}(\sigma)$ provides transformed noise level conditioning, and $c_{\text{skip}}(\sigma)$ adaptively balances signal mixing. This principled framework improves the robustness and efficiency of diffusion models, enabling state-of-the-art performance across various generative tasks.

## C.2   Multi-task Essence of World Model Learning

Modern world models [15, 7, 2, 8] typically address two fundamental prediction tasks: the modeling of environment dynamics through observations and the prediction of reward signals. The learning of these tasks requires distinct considerations based on the complexity of the environment. In simple low-dimensional settings, separate learning approaches suffice for each task. However, the introduction of high-dimensional visual inputs fundamentally alters this paradigm, as partial observability creates an inherent coupling between state estimation and reward prediction. This coupling necessitates joint learning through shared representations, an approach that aligns with established multi-task learning principles [65]. The implementation of such joint learning through shared representations introduces several technical challenges. The integration of multiple learning objectives requires careful consideration of their relative importance and interactions. A fundamental difficulty stems from the inherent scale disparity between high-dimensional visual observations and scalar reward signals. This disparity manifests in the world model learning objective, which combines observation modeling $\mathcal{L}_o(\theta)$, reward modeling $\mathcal{L}_r(\theta)$, and dynamics modeling $\mathcal{L}_d(\theta)$ losses with weights $w_o$, $w_r$, $w_d$ to control relative contributions:

$$\mathcal{L}(\theta) = w_o\mathcal{L}_o(\theta) + w_r\mathcal{L}_r(\theta) + w_d\mathcal{L}_d(\theta). \tag{11}$$

HarmonyDream [56] demonstrated that observation modeling tends to dominate this objective due to visual inputs' high dimensionality compared to scalar rewards. Their work introduced a variational formulation:

$$\mathcal{L}(\theta, w_o, w_r, w_d) = \sum_{i \in \{o,r,d\}} \mathcal{H}\left(\mathcal{L}_i(\theta), \frac{1}{w_i}\right) = \sum_{i \in \{o,r,d\}} w_i\mathcal{L}_i(\theta) + \log\left(\frac{1}{w_i}\right), \tag{12}$$

where $\mathcal{H}(\mathcal{L}_i(\theta), w_i) = w_i\mathcal{L}_i(\theta) + \log(1/w_i)$ dynamically balances the losses by maintaining $\mathbb{E}[w^* \cdot \mathcal{L}] = 1$. This harmonization technique can substantially enhance sample efficiency and performance. Our work extends these insights through the integration of dynamic task balancing mechanisms into our EDELINE world model architecture.

# D   Additional Experiments and Experiment Details

In this section, we provide extensive additional experiments and detailed experimental setups that complement the primary results presented in the main manuscript. These supplementary materials include in-depth analyses, ablation studies, and technical details that further validate and illuminate EDELINE's advantages.

## D.1   EDM Network Preconditioners and Training

Following DIAMOND [8], we use the EDM preconditioners from [64] for normalization and rescaling to improve network training:

$$c_{in}^{\tau} = \frac{1}{\sqrt{\sigma(\tau)^2 + \sigma_{data}^2}} \tag{13}$$

$$c_{out}^{\tau} = \frac{\sigma(\tau)\sigma_{data}}{\sqrt{\sigma(\tau)^2 + \sigma_{data}^2}} \tag{14}$$

$$c_{noise}^{\tau} = \frac{1}{4}\log(\sigma(\tau)) \tag{15}$$

$$c_{skip}^{\tau} = \frac{\sigma_{data}^2}{\sigma_{data}^2 + \sigma^2(\tau)}, \tag{16}$$

where $\sigma_{data} = 0.5$.

The noise parameter $\sigma(\tau)$ is sampled to maximize the effectiveness of training:

$$\log(\sigma(\tau)) \sim \mathcal{N}(P_{mean}, P_{std}^2), \tag{17}$$

where $P_{mean} = -0.4$, $P_{std} = 1.2$.

## D.2  Atari 100K Training Curves

Fig. 6 presents the detailed training curves for all individual games in the Atari100k benchmark.

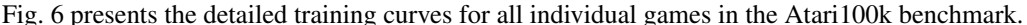

Figure 6: Training curves of EDELINE on the Atari100k benchmark for individual games (400K environment steps). The solid lines represent the average scores over 3 seeds, and the filled areas indicate the standard deviation across these 3 seeds.

### D.3 Atari 100k Qualitative Analysis

To provide deeper insights into EDELINE's superior performance, we conduct qualitative analysis on three Atari games where our approach demonstrates the most significant improvements over DIAMOND: BankHeist, DemonAttack, and Hero. Fig. 7 presents temporal sequences comparing ground truth gameplay with predictions from both EDELINE and DIAMOND world models.

In BankHeist, agents maximize scores through repeated map traversal to encounter new enemies. EDELINE's SSM-enhanced world model maintains consistent tracking of the player character position throughout prediction sequences, while DIAMOND's model shows progressive degradation with the character eventually disappearing from predictions. For DemonAttack, EDELINE successfully captures the relationship between enemy hits and score updates in its predictions. DIAMOND preserves basic visual structure but fails to reflect these crucial state transitions. The Hero environment showcases EDELINE's long-range prediction capabilities, accurately capturing sequences of the player breaking obstacles and navigating new areas. These qualitative results highlight how EDELINE's architectural innovations - SSM-based memory, unified training, and diffusion modeling - enable robust state tracking, action-consequence modeling, and temporal consistency. These capabilities directly contribute to EDELINE's superior performance on the Atari 100k benchmark.

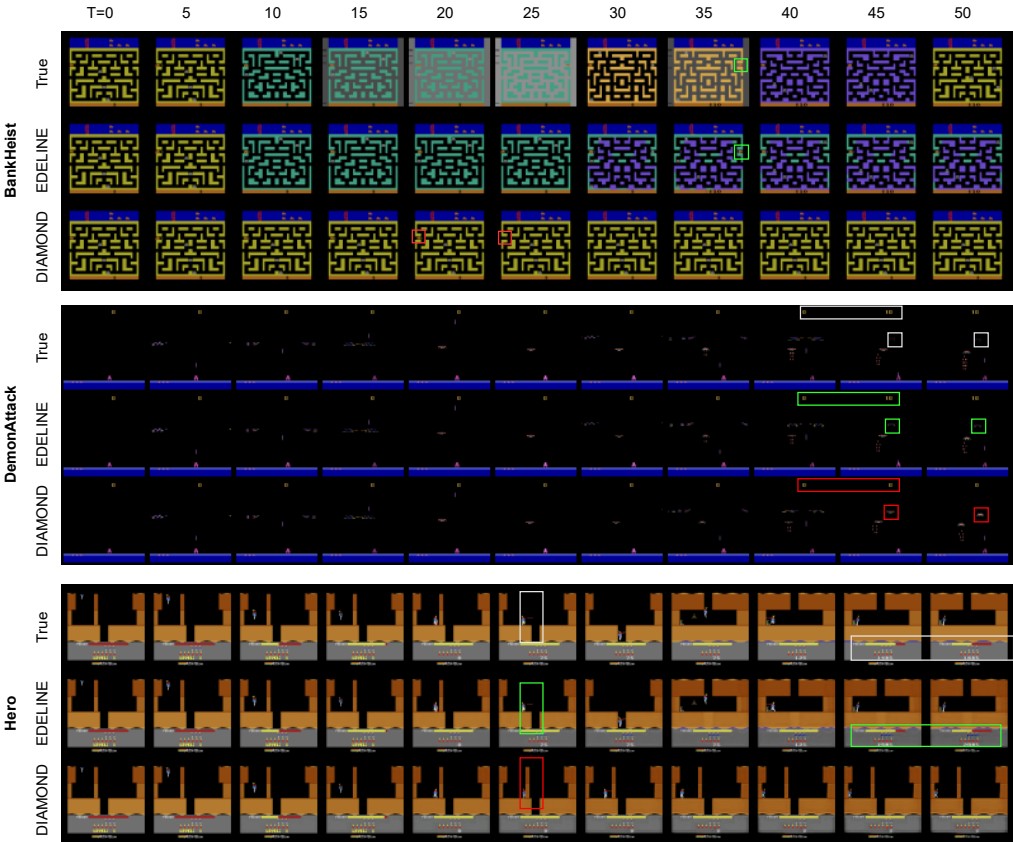

Figure 7: Qualitative comparison of world model predictions on three Atari games. Each panel shows temporal sequences comparing ground truth with EDELINE and DIAMOND predictions. In BankHeist (top), EDELINE successfully tracks the player character while DIAMOND loses this information. DemonAttack (middle) demonstrates EDELINE's accurate modeling of score updates upon successful hits. Hero (bottom) showcases EDELINE's ability to maintain consistent predictions of complex character-environment interactions across extended sequences. Colored boxes highlight successful (green) and failed (red) predictions of key game elements.

## D.4 Extended Atari 100k Benchmark Performance Analysis

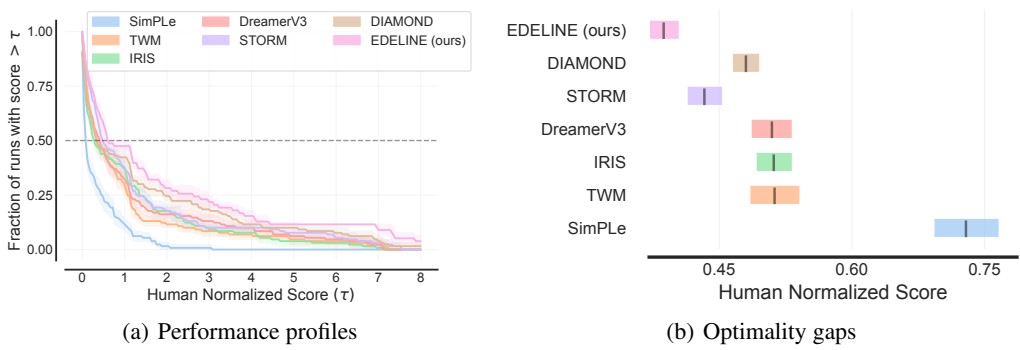

(a) Performance profiles

(b) Optimality gaps

Figure 8: Additional performance analyses. (a) Performance profiles showing fraction of runs achieving scores above different human normalized score thresholds. (b) Optimality gaps demonstrating relative distance to theoretical optimal performance across different methods.

To provide additional performance insights, we analyze EDELINE using performance profiles and optimality gaps [58]. Fig. 8(a) shows the empirical cumulative distribution of human-normalized scores (HNS) across all 26 Atari games. The y-axis indicates the fraction of games achieving scores above each HNS threshold $\tau$. EDELINE consistently maintains a higher fraction of games across different thresholds compared to baseline methods, particularly in the middle range ($\tau$ between 1 and 4). We further analyze model performance through optimality gaps shown in Fig. 8(b). The optimality gap measures the distance between model performance and theoretical optimal behavior. EDELINE achieves the smallest optimality gap among all compared methods, demonstrating its effectiveness in approaching optimal performance across the Atari 100k benchmark suite. These detailed analyses complement the aggregate metrics presented in Section 6.2 by providing a more granular view of performance distribution and theoretical efficiency.

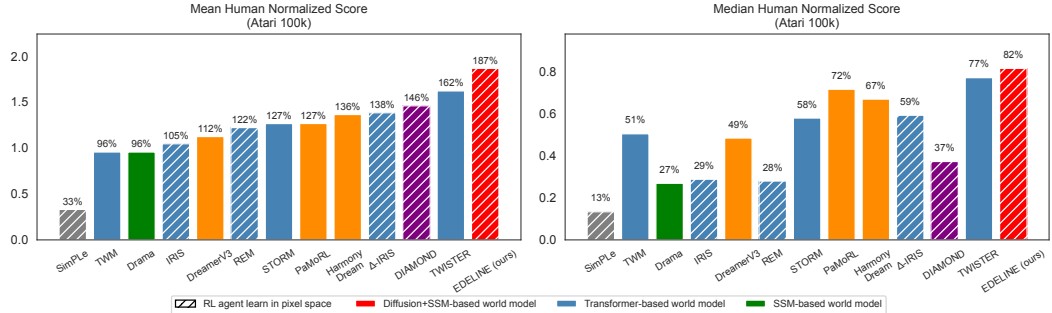

Figure 9: **Comparison of state-of-the-art model-based RL methods without using look-ahead search techniques on the Atari 100k benchmark.** EDELINE outperforms all existing model-based approaches on the Atari 100k benchmark. Previous methods can be categorized by their world model architectures: Transformer-based models (TWM [32], IRIS [35], REM [66], STORM [33], Δ-IRIS [59], and TWISTER [67]), RNN-based models (DreamerV3 [2], PaMoRL [68], and HarmonyDream [56]), SSM-based model (Drama [48]), and Diffusion-based model (DIAMOND [8]). Our proposed EDELINE advances the state-of-the-art by integrating diffusion modeling with state space models, combining their respective strengths in visual generation and temporal modeling.

### D.5 Atari 100k Linear Probing Analysis

To quantitatively evaluate the information content captured in EDELINE's hidden representations, we conducted a comprehensive linear probing analysis, which assesses how effectively the model's hidden states encode information critical for both reward prediction and observation prediction tasks.

Table 3: Linear probing loss of reward and observation prediction from EDELINE and DIAMOND hidden states. $\mathcal{L}_{\text{rew}}$ (reward loss) was measured using cross-entropy loss on reward prediction, while $\mathcal{L}_{\text{obs}}$ (observation loss) was measured using MSE loss on predicting encoded observation features. EDELINE outperforms DIAMOND in observation prediction with an average 57.3% reduction in loss, while both models achieve comparable performance on reward prediction (i.e., average $\mathcal{L}_{\text{rew}}$ of 0.0246 vs. 0.0258). These results demonstrate that EDELINE's unified hidden representation successfully encodes information required for both tasks.

| Environment | EDELINE $\mathcal{L}_{\text{rew}}$ | DIAMOND $\mathcal{L}_{\text{rew}}$ | EDELINE $\mathcal{L}_{\text{obs}}$ | DIAMOND $\mathcal{L}_{\text{obs}}$ |
|---|---|---|---|---|
| Alien | **0.0434** | 0.0663 | **0.6161** | 1.1798 |
| Amidar | **0.0186** | 0.0187 | **0.6430** | 0.7520 |
| Assault | 0.0779 | **0.0270** | 0.4894 | **0.4199** |
| Asterix | 0.0114 | **0.0018** | **0.8247** | 1.4411 |
| BankHeist | **0.0023** | 0.0039 | **0.0312** | 2.5398 |
| BattleZone | **0.0129** | 0.0220 | 0.2265 | **0.1413** |
| Boxing | 0.0482 | **0.0011** | **0.3089** | 0.5328 |
| Breakout | 0.0615 | **0.0264** | 0.6224 | **0.4871** |
| ChopperCommand | 0.0261 | **0.0113** | **0.6815** | 1.3819 |
| CrazyClimber | **0.0029** | 0.0034 | **0.2844** | 0.5513 |
| DemonAttack | **0.0609** | 0.0726 | **0.8588** | 1.0039 |
| Freeway | **0.0000** | **0.0000** | **0.2163** | 0.3911 |
| Frostbite | **0.0001** | 0.0002 | **0.1992** | 0.7435 |
| Gopher | **0.0090** | 0.0091 | **0.3139** | 0.9558 |
| Hero | **0.0065** | 0.0088 | **0.0708** | 0.1013 |
| Jamesbond | **0.0054** | 0.0058 | **0.5931** | 1.1080 |
| Kangaroo | **0.0009** | 0.0046 | **0.0992** | 0.5054 |
| Krull | 0.0728 | **0.0455** | **0.3216** | 0.8615 |
| KungFuMaster | 0.0048 | **0.0043** | **0.2674** | 0.8735 |
| MsPacman | **0.0044** | 0.0669 | **0.2510** | 0.8769 |
| Pong | **0.0000** | **0.0000** | **0.1770** | 1.3162 |
| PrivateEye | **0.0014** | 0.0144 | **0.1188** | 0.4931 |
| Qbert | 0.0240 | **0.0025** | **0.1813** | 0.2530 |
| RoadRunner | 0.0658 | **0.0310** | **0.2032** | 0.7896 |
| Seaquest | **0.0008** | 0.0299 | **0.2159** | 0.7852 |
| UpNDown | **0.1012** | 0.2110 | **0.5967** | 1.0783 |
| ViZDoom-DeadlyCorridor | **0.0008** | 0.0094 | **0.3354** | 1.2848 |
| Average Loss | **0.0246** | 0.0258 | **0.3610** | 0.8462 |

We trained linear probes on the hidden states extracted from both EDELINE and DIAMOND models. For our experimental setup, we collected 50 trajectories as training data and 10 trajectories as validation data for each environment, with five independent random seeds employed to ensure statistical robustness. For reward prediction, we trained a single linear layer to predict rewards for 50 epochs. For observation feature prediction, we trained a multi-layer perceptron (MLP) to predict encoded features of the subsequent observation (extracted from the pre-trained actor-critic network encoder) for 100 epochs. For EDELINE, we utilized the Mamba hidden states, while for DIAMOND, we employed the LSTM hidden states from its reward-termination model. We conducted these experiments across 26 Atari environments and the challenging ViZDoom-DeadlyCorridor environment to ensure comprehensive evaluation.

Table 3 presents the results of our linear probing analysis. EDELINE and DIAMOND demonstrate comparable performance on reward prediction, with average losses of 0.0246 and 0.0258 respectively. EDELINE outperforms DIAMOND in 15 out of 27 environments on this task. More significantly, EDELINE substantially outperforms DIAMOND in observation prediction, with an average $57.3\%$ reduction in loss. EDELINE achieves lower observation prediction loss in 24 out of 27 environments. The substantial improvement in observation prediction while maintaining comparable reward prediction performance demonstrates that EDELINE's unified hidden representation successfully captures information required for both tasks.

## D.6 Generation Quality Evaluation

To quantitatively assess the generation quality of our proposed world model, we conducted a comprehensive evaluation measuring pixel-wise Mean Squared Error (MSE) across 26 Atari environments and the challenging ViZDoom-DeadlyCorridor environment. This analysis provides direct evidence of EDELINE's improved predictive accuracy compared to DIAMOND.

Table 4: Pixel-wise MSE comparison between EDELINE and DIAMOND across 27 environments. Lower values (in **bold**) indicate better performance. The rightmost column shows the imagination horizon length for each environment. The bottom row reports the average normalized score (EDELINE MSE / DIAMOND MSE), with values below 1.0 indicating EDELINE's overall superior performance.

| Environment | EDELINE | DIAMOND | Imagine Horizon Length |
|---|---|---|---|
| Alien | **0.0063** | 0.0064 | 635 |
| Amidar | **0.0177** | 0.0235 | 1029 |
| Assault | **0.1011** | 0.2528 | 566 |
| Asterix | 0.0190 | **0.0177** | 1493 |
| BankHeist | **0.1510** | 0.1733 | 2019 |
| BattleZone | 0.0306 | **0.0292** | 1407 |
| Boxing | **0.0068** | 0.0174 | 1371 |
| Breakout | 0.0420 | **0.0415** | 1796 |
| ChopperCommand | 0.0084 | **0.0082** | 3006 |
| CrazyClimber | **0.0666** | 0.0686 | 3219 |
| DemonAttack | **0.0312** | 0.0318 | 1474 |
| Freeway | **0.0015** | 0.0030 | 1986 |
| Frostbite | **0.0338** | 0.0370 | 564 |
| Gopher | **0.0152** | 0.0172 | 2689 |
| Hero | **0.1559** | 0.2005 | 2103 |
| Jamesbond | **0.2967** | 0.3023 | 2212 |
| Kangaroo | **0.0061** | 0.0063 | 3241 |
| Krull | 0.1577 | **0.1575** | 1326 |
| KungFuMaster | 0.0210 | 0.0210 | 1864 |
| MsPacman | **0.0179** | 0.0197 | 892 |
| Pong | 0.0035 | **0.0034** | 1532 |
| PrivateEye | 0.1043 | **0.0638** | 2454 |
| Qbert | **0.0435** | 0.0460 | 1434 |
| RoadRunner | **0.0526** | 0.0532 | 960 |
| Seaquest | **0.0107** | 0.0136 | 1810 |
| UpNDown | **0.1223** | 0.1234 | 1279 |
| ViZDoom-DeadlyCorridor | **0.0179** | 0.0184 | 73 |
| Mean Normalized Score | **0.918** | 1.000 | |

Table 4 presents the comprehensive results of our pixel-wise MSE comparison. The analysis reveals that EDELINE achieves lower MSE than DIAMOND in 20 out of 27 environments. For meaningful comparison across environments with varying visual complexity, we calculated the average normalized score (EDELINE MSE / DIAMOND MSE) across all environments. Our analysis demonstrates that EDELINE achieves an average normalized score of 0.918, which represents an overall $8.2\%$ reduction in pixel-level error compared to DIAMOND.

The superior generation quality of EDELINE can be attributed to its utilization of Mamba to incorporate longer observation history beyond the four frames employed by DIAMOND. This capability enables EDELINE to maintain more consistent predictions over extended horizons, which proves particularly advantageous in environments that necessitate memory (such as ViZDoom) or involve complex dynamics.

## D.7 Training Time Profile

We performed detailed training time profiling of EDELINE to evaluate its computational efficiency compared to DIAMOND. Table 5 provides a comprehensive breakdown of training time components across different scales, while Table 6 directly compares EDELINE with DIAMOND. Our profiling reveals that EDELINE achieves notable efficiency improvements in world model training. For world model updates, EDELINE is approximately 26.8% faster than DIAMOND. This efficiency gain stems from our unified architecture approach. While DIAMOND employs a two-stage training process (diffusion model for observations plus a separate CNN-LSTM network for reward/termination prediction), EDELINE's unified architecture enables joint learning of observations, rewards, and terminations through shared representations. In addition, Mamba's parallel scan algorithm contributes to this computational advantage during training. For actor-critic updates, EDELINE requires approximately 17.7% more computation time than DIAMOND. This difference occurs because while both methods use identical actor-critic architectures, EDELINE's world model involves more complex inference during imagined rollouts due to SSM processing and cross-attention mechanisms. Specifically, as shown in Table 5, each imagination step requires 24.4ms, with 17.2ms dedicated to observation prediction and 6.5ms to reward/termination prediction. Overall, these timing differences balance out, resulting in comparable total training time between the two approaches. This demonstrates that EDELINE successfully maintains computational efficiency comparable to DIAMOND while delivering significantly enhanced memory capabilities and performance.

Table 5: Detailed breakdown of training time components across different scales. Profiling performed using a Nvidia RTX 4090 with default hyperparameters. Measurements are representative, as exact durations depend on specific hardware, environment, and training stage.

| Single update | Time (ms) | Detail (ms) |
|---|---|---|
| Total | 548.6 | 148.6 + 400 |
| World model update | 148.6 | - |
| Actor-Critic model update | 400 | $15 \times 24.4 + 34$ |
| Imagination step (x 15) | 24.4 | 17.2 + 6.5 + 0.7 |
| Next observation prediction | 17.2 | - |
| Denoising step (x 3) | 5.7 | - |
| Reward/Termination prediction | 6.5 | - |
| Action prediction | 0.7 | - |
| Loss computation and backward | 34 | - |

| Epoch | Time (s) | Detail (s) |
|---|---|---|
| Total | 219.4 | 59.4 + 160 |
| World model | 59.4 | $400 \times 148.6 \times 10^{-3}$ |
| Actor-Critic model | 160 | $400 \times 400 \times 10^{-3}$ |

| Run | Time (days) | Detail (days) |
|---|---|---|
| Total | 2.9 | 2.5 + 0.4 |
| Training time | 2.5 | $1000 \times 219.4 / (24 \times 3600)$ |
| Other (collection, evaluation, checkpointing) | 0.4 | - |

Table 6: Computational efficiency comparison between DIAMOND and EDELINE across different training stages, showing per-stage timing and relative differences.

| Module | DIAMOND (ms/s/days) | EDELINE (ms/s/days) | Difference (%) |
|---|---|---|---|
| **Single Update** | | | |
| Total | 543 ms | 548.6 ms | +1.03% |
| World Model Update | 203 ms | 148.6 ms | -26.80% |
| Actor-Critic Model Update | 340 ms | 400 ms | +17.65% |
| **Epoch** | | | |
| Total | 217 s | 219.4 s | +1.11% |
| World Model | 81 s | 59.4 s | -26.67% |
| Actor-Critic Model | 136 s | 160 s | +17.65% |

## D.8 Ablation Studies

To systematically validate the effectiveness of EDELINE's key architectural components, we perform comprehensive ablation studies across multiple environments. These studies isolate the contribution of each design decision and demonstrate their impact on model performance. The ablation studies in Appendix D.8.1 and D.8.2 focus on five representative environments for validating the proposed key components. These include four Atari games where EDELINE demonstrates significant improvements over DIAMOND (BankHeist, DemonAttack, Hero, Seaquest), and MiniGrid-MemoryS9 for memory capability evaluation. This selection provides comprehensive validation across visual prediction quality and memorization requirements. To compare Transformer and Mamba architectures for the Recurrent Embedding Module, we conducted experiments on memory tasks such as MiniGrid MemoryS7/S9 and Crafter. These results are presented in Appendix D.8.4.

### D.8.1 Choice of REM architecture

To validate the selection of Mamba for REM, we compare its performance against traditional linear-time sequence models GRU and LSTM across five environments. Fig. 10 illustrates that although all models achieve reasonable performance, Mamba demonstrates more stable learning curves and superior final performance, particularly in memory-intensive tasks such as BankHeist and MiniGrid-MemoryS9. GRU and LSTM models exhibit increased training variance with lower final scores, which validates Mamba's effectiveness.

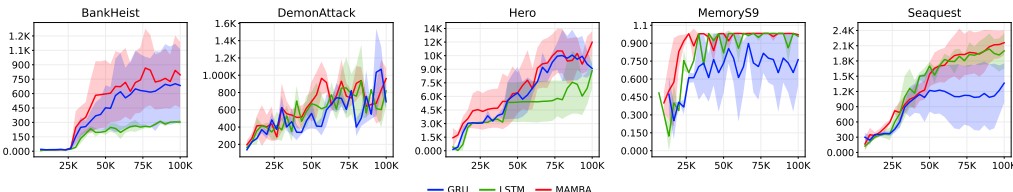

Figure 10: Performance comparison of different linear-time sequence models as REM architecture across five environments. Training curves show mean and standard deviation over three seeds. Mamba (red) shows more stable training progression and superior final performance compared to GRU (blue) and LSTM (green).

### D.8.2 Cross-Attention in Next-Frame Predictor

To evaluate whether cross-attention improves the Next-Frame Predictor's ability to process information-rich hidden embeddings, we examine EDELINE with and without this mechanism. As depicted in Fig. 11, the original EDELINE demonstrates superior performance in BankHeist, MemoryS9, and Seaquest where rich contextual information processing proves essential. The MemoryS9 environment validates this necessity, as models must integrate historical information for complete representation reconstruction. Cross-attention enables effective fusion of hidden embeddings with visual features for temporal context integration.

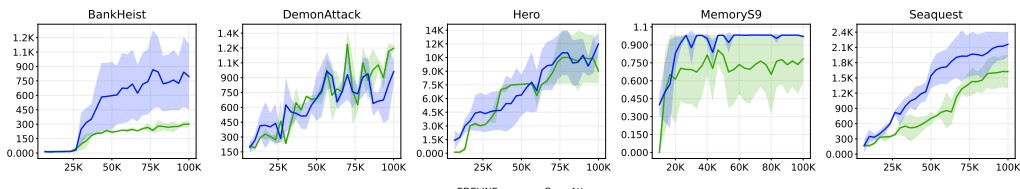

Figure 11: Ablation study comparing EDELINE with cross-attention blocks (blue) and without (green) across five test environments. Training curves depict mean and standard deviation over three seeds. EDELINE's cross-attention mechanism provides advantages in environments requiring rich contextual information processing.

### D.8.3 Effect of Harmonizers

To validate the effectiveness of harmonizers, we evaluate performance across the full Atari 100k benchmark.

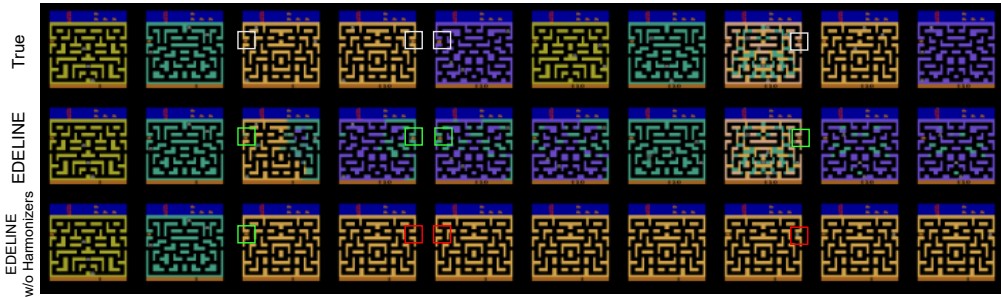

(a) BankHeist

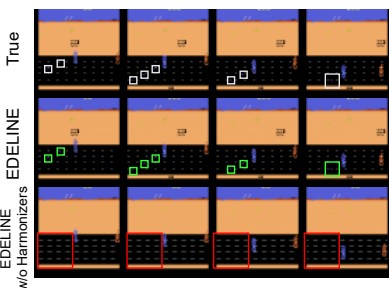

(b) RoadRunner

Figure 12: Effect of harmonizers on world model predictions. (a) Without harmonizers, EDELINE fails to maintain consistent character modeling in BankHeist. (b) In RoadRunner, removal of harmonizers leads to loss of reward-relevant visual details. Colored boxes highlight successful (green) and failed (red) predictions of key game elements.

Table 7 presents quantitative performance comparison between EDELINE with and without harmonizers. The ablation results demonstrate significant performance degradation across multiple environments when harmonizers are removed, with particularly notable drops in games requiring precise reward related visual detail retention (BankHeist and RoadRunner). Qualitative analysis on these two environments, which showed the largest performance improvements with harmonizers, reveals how harmonizers contribute to world model performance. In BankHeist (Fig. 12(a)), removing harmonizers causes the world model to lose track of game characters, failing to maintain consistent agent representation across prediction sequences. Similarly, in RoadRunner (Fig. 12(b)), the model without harmonizers fails to capture reward-relevant visual details, degrading its ability to model critical game state information.

These results demonstrate how harmonizers help achieve a crucial balance between observation modeling and reward prediction. Without harmonizers, the world model struggles with fine-grained task-relevant observations in both environments - tracking small character sprites in BankHeist and capturing critical game state details in RoadRunner. By maintaining this dynamic equilibrium between observation and reward modeling, harmonizers enable EDELINE to effectively learn compact task-centric dynamics while preserving essential visual details for sample-efficient learning.

Table 7: To evaluate the benefits of Harmonizers in detail, we compare DreamerV3 and its variant with Harmonizers (HarmonyDream), alongside EDELINE without Harmonizers and the complete EDELINE on the 26 games in the Atari 100k benchmark. Bold numbers indicate the highest scores.

| Game | Random | Human | DreamerV3 | HarmonyDream | EDELINE w/o Harmonizers | EDELINE (ours) |
|---|---|---|---|---|---|---|
| Alien | 227.8 | 7127.7 | 959.4 | 889.7 | **1086.2** | 974.6 |
| Amidar | 5.8 | 1719.5 | 139.1 | 141.1 | 212.5 | **299.5** |
| Assault | 222.4 | 742.0 | 705.6 | 1002.7 | 1180.1 | **1225.8** |
| Asterix | 210.0 | 8503.3 | 932.5 | 1140.3 | **4612.8** | 4224.5 |
| BankHeist | 14.2 | 753.1 | 648.7 | **1068.6** | 139.8 | 854.0 |
| BattleZone | 2360.0 | 37187.5 | 12250.0 | **16456.0** | 2685.0 | 5683.3 |
| Boxing | 0.1 | 12.1 | 78.0 | 79.6 | **88.3** | 88.1 |
| Breakout | 1.7 | 30.5 | 31.1 | 52.6 | **255.7** | 250.5 |
| ChopperCommand | 811.0 | 7387.8 | 410.0 | 1509.6 | **2616.5** | 2047.3 |
| CrazyClimber | 10780.5 | 35829.4 | 97190.0 | 82739.0 | 96889.5 | **101781.0** |
| DemonAttack | 152.1 | 1971.0 | 303.3 | 202.6 | 924.2 | **1016.1** |
| Freeway | 0.0 | 29.6 | 0.0 | 0.0 | 33.8 | **33.8** |
| Frostbite | 65.2 | 4334.7 | **909.4** | 678.7 | 289.0 | 286.8 |
| Gopher | 257.6 | 2412.5 | 3730.0 | **13042.8** | 7982.7 | 6102.3 |
| Hero | 1027.0 | 30826.4 | 11160.5 | **13378.0** | 9366.2 | 12780.8 |
| Jamesbond | 29.0 | 302.8 | 444.6 | 317.1 | 527.5 | **784.3** |
| Kangaroo | 52.0 | 3035.0 | 4098.3 | 5117.6 | 3970.0 | **5270.0** |
| Krull | 1598.0 | 2665.5 | 7781.5 | 7753.6 | 8762.7 | **9748.8** |
| KungFuMaster | 258.5 | 22736.3 | 21420.0 | 22274.0 | 14088.5 | **31448.0** |
| MsPacman | 307.3 | 6951.6 | 1326.9 | 1680.7 | 1773.3 | **1849.3** |
| Pong | -20.7 | 14.6 | 18.4 | 18.6 | 18.4 | **20.5** |
| PrivateEye | 24.9 | 69571.3 | 881.6 | **2932.2** | 73.0 | 99.5 |
| Qbert | 163.9 | 13455.0 | 3405.1 | 3932.5 | 5062.3 | **6776.2** |
| RoadRunner | 11.5 | 7845.0 | 15565.0 | 14646.4 | 23272.5 | **32020.0** |
| Seaquest | 68.4 | 42054.7 | 618.0 | 665.3 | 1277.2 | **2140.1** |
| UpNDown | 533.4 | 11693.2 | 7567.1 | **10873.6** | 2844.6 | 5650.3 |
| #Superhuman (↑) | 0 | N/A | 9 | 11 | 11 | **13** |
| Mean (↑) | 0.000 | 1.000 | 1.124 | 1.364 | 1.674 | **1.866** |

Table 8: Performance comparison between Transformer and Mamba-based REM architectures across memory-demanding environments. Values for MiniGrid environments represent success rates, while Crafter values show average return over three independent seeds after 1M environment steps.

| REM Architecture | MiniGrid-MemoryS7 | MiniGrid-MemoryS9 | Crafter | REM Inference FLOPs |
|---|---|---|---|---|
| Transformer | 0.980 | 0.978 | $10.2 \pm 0.8$ | 263.484M |
| Mamba | 0.981 | 0.982 | **$11.5 \pm 0.9$** | **213.307M** |

### D.8.4 Transformer-based Recurrent Embedding Module

To investigate the efficacy of Mamba compared to Transformer architecture, we implemented a Transformer-based variant of the Recurrent Embedding Module (REM). This ablation study aims to assess both performance and computational efficiency across memory-demanding environments, which provides critical insights into architecture selection for world modeling. We evaluated both architectures on the MiniGrid-MemoryS7, MiniGrid-MemoryS9, and Crafter environments, which specifically challenge a model's ability to retain and utilize historical information.

As demonstrated in Table 8, both Transformer and Mamba architectures achieve comparable performance across the evaluated memory-demanding environments. On the MiniGrid-Memory tasks, both architectures attain success rates approaching optimal performance (>0.97). In the more complex Crafter environment, which requires nuanced long-term memory and planning capabilities, Transformer achieves an average return of $10.2 \pm 0.8$ while Mamba achieves $11.5 \pm 0.9$, demonstrating that both architectures can effectively model long-term dependencies in world modeling contexts.

From a computational perspective, the REM inference FLOPs comparison reveals that Mamba requires 213.307M FLOPs compared to Transformer's 263.484M FLOPs, representing approximately 19% lower computational cost attributed to its linear time complexity. These results substantiate that both Transformer and Mamba represent viable architectural alternatives for the REM component.

Our primary objective is to addresses the memory limitations of existing approaches. Our selection of Mamba is motivated by its simplicity and linear-time complexity, which achieves satisfactory performance while maintaining computational efficiency. Further exploration of superior architectural designs that enhance both efficiency and accuracy for diffusion-based world modeling constitutes an important direction for future research.

### D.8.5 Frame Stacking Ablation

To validate the necessity of EDELINE's frame-stacking design, we investigate whether both SSM hidden embeddings and explicit frame stacking are essential for achieving optimal performance. Our architectural design maintains both components based on the hypothesis that they serve complementary roles in the diffusion process: the hidden embedding captures long-term temporal dependencies and global context across the entire sequence history, while the explicit frame stacking provides pixel-based spatial details from recent observations that are crucial for pixel-level prediction accuracy. The diffusion model benefits from both the abstract temporal representations (via cross-attention with hidden embedding) and direct access to recent visual features (via frame stacking) for generating high-fidelity predictions.

To empirically validate this design choice, we conducted an ablation study comparing three configurations: (1) EDELINE with both components, (2) EDELINE with hidden embedding only (without frame stacking), and (3) DIAMOND as a baseline. We evaluated these variants on two environments with contrasting characteristics: Atari Breakout, which requires minimal long-term memory while demanding high visual detail processing, and MiniGrid-MemoryS9, which requires substantial memory capabilities while having simpler visual complexity.

Table 9: Frame Stacking Ablation Study. Performance comparison demonstrating the necessity of both SSM hidden states and explicit frame stacking across environments with different memory and visual complexity requirements.

| Environment | DIAMOND | EDELINE w/o frame-stack | EDELINE |
|---|---|---|---|
| MiniGrid-MemoryS9 | 0.380 | **0.985** | **0.982** |
| Atari Breakout | 132.5 | 145.6 | **250.5** |

As shown in Table 9, the results demonstrate distinct patterns across the two environments. In Atari Breakout, EDELINE without frame stacking achieves a score of 145.6, which represents an improvement over DIAMOND's 132.5 but remains substantially below the full EDELINE's 250.5. This significant performance gap in a visually complex environment highlights the critical importance of explicit frame stacking for preserving fine-grained visual details necessary for accurate pixel-level prediction. In contrast, MiniGrid-MemoryS9 presents a different pattern: EDELINE without frame stacking achieves a success rate of 0.985, which is comparable to the full EDELINE's 0.982 and substantially outperforms DIAMOND's 0.380. This near-optimal performance in a memory-intensive but visually simpler environment validates that the hidden embedding alone can effectively capture the temporal dependencies required for success when visual complexity is limited.

These results substantiate our architectural design by demonstrating that the two components serve complementary and essential roles. The hidden embedding provides the long-term memory capacity necessary for temporal reasoning, as evidenced by the strong performance in MiniGrid-MemoryS9 even without frame stacking. Conversely, the explicit frame stacking preserves the high-fidelity visual information crucial for environments with rich visual complexity, as demonstrated by the substantial performance improvement in Atari Breakout when both components are present. This ablation study validates that our dual input design is not redundant but rather represents a deliberate architectural choice that enables EDELINE to excel across diverse environments with varying memory and visual complexity requirements.

### D.9 ViZDoom Environment Specifications

This appendix provides detailed specifications for the ViZDoom scenarios used in our experiments. We evaluate EDELINE on five key scenarios that test different agent capabilities in first-person 3D environments.

#### D.9.1 DeadlyCorridor

- **Objective**: Navigate through enemy fire to acquire armor at corridor end
- **Reward Mechanism**:
    - +1 for each enemy killed
    - +1 for armor acquisition
    - -0.01 for each damage instance received
    - -0.01 move backward
- **Evaluation Metric**: Binary success if armor acquired before episode termination
- **Episode Termination**: Agent death or 512 environment interaction steps.

#### D.9.2 HealthGathering

- **Objective**: Survive by collecting medkits in toxic environment
- **Reward Mechanism**:
    - +1 for each medkit collected
- **Evaluation Metric**: Final health percentage (0-100%)
- **Episode Termination**: Agent death or 512 environment interaction steps.

#### D.9.3 PredictPosition

- **Objective**: Anticipate enemy movement to land delayed rocket hit
- **Reward Mechanism**:
    - +1 for successful enemy kill
- **Evaluation Metric**: Binary success if enemy eliminated
- **Episode Termination**: Successful kill or 75 environment interaction steps.

#### D.9.4 Basic

- **Objective**: Eliminate stationary enemy with limited ammunition
- **Reward Mechanism**:
    - +1 for successful kill
- **Evaluation Metric**: Binary success if enemy eliminated
- **Episode Termination**: Successful kill or 75 environment interaction steps.

#### D.9.5 DefendCenter

- **Objective**: Survive against infinite enemy waves
- **Reward Mechanism**:
    - +1 per enemy killed
    - -1 for agent death
- **Evaluation Metric**: Total enemies killed per episode
- **Episode Termination**: Agent death or 512 environment interaction steps.

All scenarios use the following common configuration parameters unless otherwise specified:

- Observation space: (64,64,3)
- Action space: Discrete movement and attack actions

## D.10 Crafter Experiments Qualitative Results

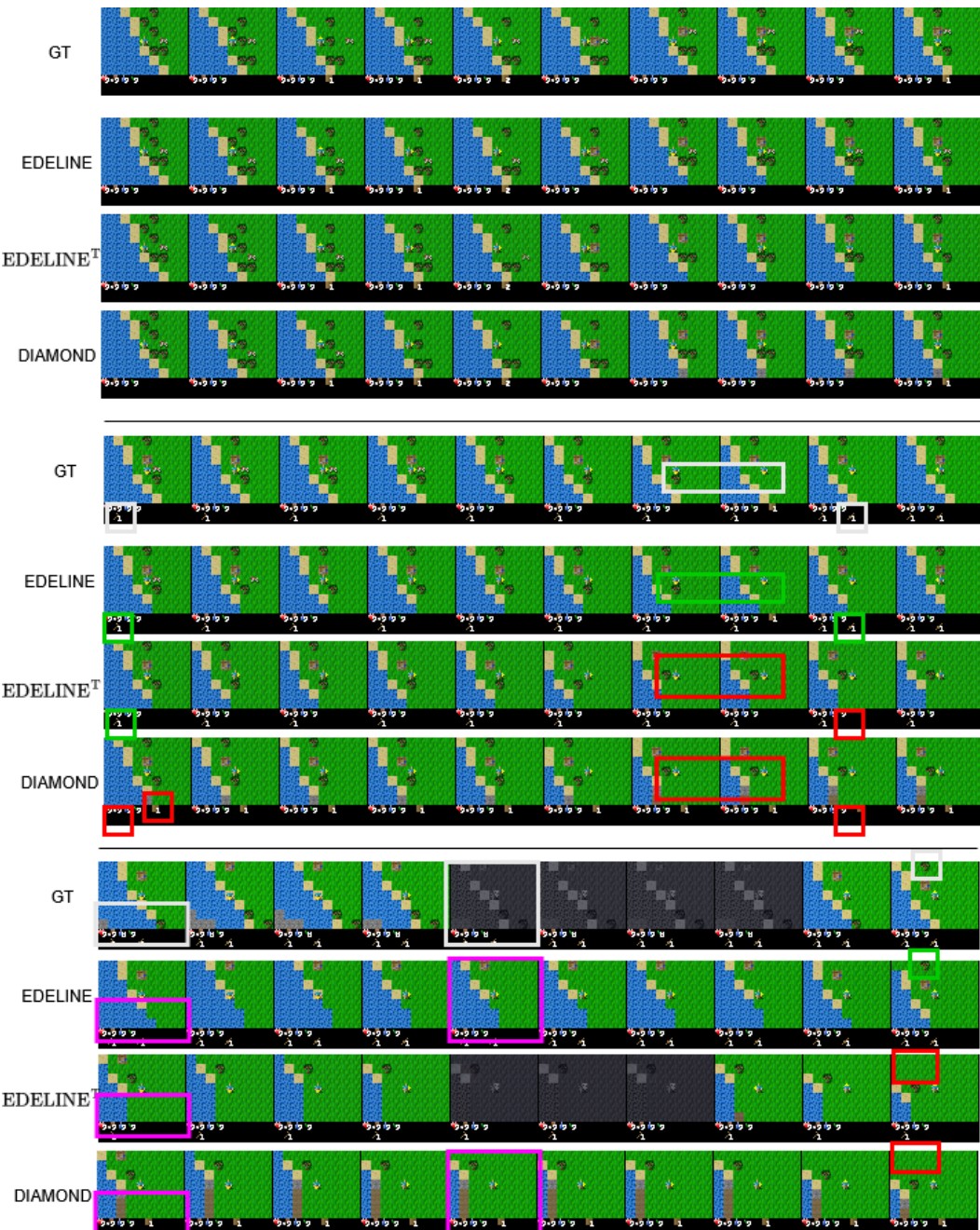

Figure 13: Qualitative comparison of world model predictions in Crafter. We compare GT (Ground Truth), EDELINE, EDELINE$^T$ (Transformer-based variant), and DIAMOND. The red boxes indicate prediction errors, the pink boxes show acceptable errors, and the green boxes highlight correct predictions of key elements.

In Fig. 13, we can observe comparative results between the different model architectures. In timesteps 1-10 (top row), all three methods demonstrate reasonable consistency in their predictions. Starting from timesteps 11-20 (middle row), notable differences emerge: DIAMOND fails to correctly represent crafted materials, and both DIAMOND and EDELINE$^T$ incorrectly predict outcomes when the player chops trees and crafts new materials. EDELINE maintains higher prediction accuracy throughout these interactions. In timesteps 21-30 (bottom row), all methods generate acceptable

predictions for previously unseen regions. While these differ from Ground Truth, they remain acceptable as they preserve gameplay-critical elements. During sleep actions, both EDELINE and DIAMOND fail to predict the sleeping state, which we classify as acceptable error given that sleep is a random event triggered independently of player actions. Most importantly, when the player returns to previously visited areas, only EDELINE successfully remembers and reproduces critical environmental features such as trees. EDELINE$^T$ and DIAMOND both fail to recall the crafting table and trees outside the map border. This evidence demonstrates EDELINE's superior memorization capability for maintaining environmental consistency over long horizons.

## D.11 Hyperparameters

Table 10: Hyperparameters for EDELINE.

| Hyperparameter | Symbol | Value |
|---|---|---|
| **General** | | |
| Number of epochs | — | 1000 |
| Training steps per epoch | — | 400 |
| Environment steps per epoch | — | 100 |
| Batch size | — | 32 |
| Sequence Length | $T$ | 19 |
| Epsilon (greedy) for collection | $\epsilon$ | 0.01 |
| Observation shape | (h,w,c) | (64,64,3) |
| **Actor Critic** | | |
| Imagination horizon | $H$ | 15 |
| Discount factor | $\gamma$ | 0.985 |
| Entropy weight | $\eta$ | 0.001 |
| $\lambda$-returns coefficient | $\lambda$ | 0.95 |
| LSTM dimension | — | 512 |
| Residual blocks layers | — | [1,1,1,1] |
| Residual blocks channels | — | [32,32,64,64] |
| **World Model** | | |
| Number of conditioning observations | $L$ | 4 |
| Burn-in length | $\mathcal{B}$ | 4 |
| Hidden embedding dimension | d_model | 512 |
| Reward / Termination Model Hidden Units | — | 512 |
| **Mamba** | | |
| Mamba layers | n_layers | 3 |
| Mamba state dimension | d_state | 16 |
| Expand factor | — | 2 |
| 1D Convolution Dimension | d_conv | 4 |
| Residual blocks layers | — | [1,1,1,1] |
| Residual blocks channels | — | [64,64,64,64] |
| Action embedding dimenstion | — | 128 |
| **Diffusion** | | |
| Sampling method | — | Euler |
| Number of denoising steps | — | 3 |
| Condition embedding dimension | — | 256 |
| Residual blocks layers | — | [2,2,2,2] |
| Residual blocks channels | — | [64,64,64,64] |
| **Optimization** | | |
| Optimizer | — | AdamW |
| Learning rate | $\alpha$ | 1e-4 |
| Epsilon | — | 1e-8 |
| Weight decay (World Model) | — | 1e-2 |
| Weight decay (Actor-Critic) | — | 0 |
| Max grad norm (World Model) | — | 1.0 |
| Max grad norm (Actor-Critic) | — | 100.0 |

# E  EDELINE Algorithm

We summarize the overall training procedure of EDELINE in Algorithm 1 below, which is modified from Algorithm 1 in [8]. We denote as $\mathcal{D}$ the replay dataset where the agent stores data collected from the real environment, and other notations are introduced in previous sections or are self-explanatory.

---

**Algorithm 1:** EDELINE

---

**Procedure** `training_loop()`:

  **for** *epochs* **do**

    `collect_experience(`*steps_collect*`)`

    **for** *steps_world_model* **do**

      `update_world_model()`

    **for** *steps_actor_critic* **do**

      `update_actor_critic()`

**Procedure** `collect_experience(`$n$`)`:

  $o_0^0 \leftarrow$ `env.reset()`

  **for** $t = 0$ **to** $n - 1$ **do**

    Sample $a_t \sim \pi_\theta(a_t \mid o_t^0)$

    $o_{t+1}^0, r_t, d_t \leftarrow$ `env.step(`$a_t$`)`

    $\mathcal{D} \leftarrow \mathcal{D} \cup \{o_t^0, a_t, r_t, d_t\}$

    **if** $d_t = 1$ **then**

      $o_{t+1}^0 \leftarrow$ `env.reset()`

**Procedure** `update_world_model()`:

  Sample indexes $\mathcal{I} := \{t, \ldots, t + T - 1\}$      `// sequence length` $T$

  Sample sequence $(o_i^0, a_i, r_i, d_i)_{i \in \mathcal{I}} \sim \mathcal{D}$

  Initialize $h_{t-1}$        `// MAMBA hidden state`

  **Parallel for** $i \in \mathcal{I}$ **do**

    $h_i \leftarrow f_\phi(o_i^0, a_i, h_{i-1})$ `// processed in parallel via MAMBA parallel scan`

    $\hat{r}_i \sim p_\phi(\hat{r}_i | h_i)$

    $\hat{d}_i \sim p_\phi(\hat{d}_i | h_i)$

  Compute $\mathcal{L}_{\text{rew}}(\phi) = \sum_{i \in \mathcal{I}} \text{CE}(\hat{r}_i, r_i)$      `// CE: cross-entropy loss`

  Compute $\mathcal{L}_{\text{end}}(\phi) = \sum_{i \in \mathcal{I}} \text{CE}(\hat{d}_i, d_i)$      `// CE: cross-entropy loss`

  Sample index $j \sim \text{Uniform}\{t + \mathcal{B}, \ldots, t + T - 1\}$      `// burn-in` $\mathcal{B}$ `steps`

  Sample $\log(\sigma) \sim \mathcal{N}(P_{\text{mean}}, P_{\text{std}}^2)$      `// log-normal sampling from EDM`

  Define $\tau := \sigma$      `// identity schedule from EDM`

  Sample $o_j^\tau \sim \mathcal{N}(o_j^0, \sigma^2 \mathbf{I})$      `// add Gaussian noise`

  Compute $\hat{o}_j^0 = D_\phi(o_j^\tau, \tau, o_{j-L}^0, \ldots, o_{j-1}^0, h_{j-1})$

  Compute observation modeling loss $\mathcal{L}_{\text{obs}}(\phi) = \|\hat{o}_j^0 - o_j^0\|^2$

  Update $\phi$ according to Eq. (5)

**Procedure** `update_actor_critic()`:

  Sample initial buffer $(o_{t-\mathcal{B}+1}^0, a_{t-\mathcal{B}+1}, \ldots, o_t^0) \sim \mathcal{D}$

  `// Burn in LSTM states` $\pi_\theta$`,` $V_\theta$ `and MAMBA states` $f_\phi$ `with the buffer`

  **for** $i = t$ **to** $t + H - 1$ **do**

    Sample $a_i \sim \pi_\theta(a_i \mid o_i^0)$

    Compute $h_i \leftarrow f_\phi(o_i, a_i, h_{i-1})$

    Sample reward $r_i$, next observation $o_{i+1}^0$, and termination $d_i$ via $p_\phi$

  Compute $V_\theta(o_i^0)$ for $i = t, \ldots, t + H$

  Compute RL losses $\mathcal{L}_V(\theta)$ and $\mathcal{L}_\pi(\theta)$

  Update $\pi_\theta$ and $V_\theta$

---

# F  Actor-Critic Learning Objectives

We follow DIAMOND [8] in the design of our agent behavior learning. Let $o_t$, $r_t$, and $d_t$ denote the observations, rewards, and boolean episode terminations predicted by our world model. We denote $H$ as the imagination horizon, $V_\theta$ as the value network, $\pi_\theta$ as the policy network, and $a_t$ as the actions taken by the policy within the world model.

For value network training, we use $\lambda$-returns to balance bias and variance in the regression target. Given an imagined trajectory of length $H$, we define the $\lambda$-return recursively:

$$\Lambda_t = \begin{cases} r_t + \gamma(1 - d_t)[(1 - \lambda)V_\theta(o_{t+1}) + \lambda\Lambda_{t+1}] & \text{if } t < H \\ V_\theta(o_H) & \text{if } t = H. \end{cases} \tag{18}$$

The value network $V_\theta$ is trained to minimize $\mathcal{L}_V(\theta)$, the expected squared difference with $\lambda$-returns over imagined trajectories:

$$\mathcal{L}_V(\theta) = \mathbb{E}_{\pi_\theta}\left[\sum_{t=0}^{H-1}(V_\theta(o_t) - \text{sg}(\Lambda_t))^2\right], \tag{19}$$

where $\text{sg}(\cdot)$ denotes the gradient stopping operation, following standard practice [2, 35].

For policy training, we leverage the ability to generate large amounts of on-policy trajectories in imagination using a REINFORCE objective [69]. The policy is trained to minimize:

$$\mathcal{L}_\pi(\theta) = -\mathbb{E}_{\pi_\theta}\left[\sum_{t=0}^{H-1}\log(\pi_\theta(a_t|o_{\leq t}))\text{sg}(\Lambda_t - V_\theta(o_t)) + \eta\mathcal{H}(\pi_\theta(a_t|o_{\leq t}))\right], \tag{20}$$

where $V_\theta(o_t)$ serves as a baseline to reduce gradient variance, and the entropy term $\mathcal{H}$ with weight $\eta$ encourages sufficient exploration.

