# OpenReview forum: "EDELINE: Enhancing Memory in Diffusion-based World Models via Linear-Time Sequence Modeling"
_NeurIPS.cc/2025/Conference — NeurIPS 2025 spotlight_

### Official Review · Reviewer_t2dk · 2025-06-13

**Clarity:** 4
**Significance:** 4
**Originality:** 4
**Rating:** 5
**Confidence:** 4

**Summary:**

This paper proposes EDELINE, a diffusion-based world model that incorporates long-term memory through State Space Models (SSMs). The authors address a key limitation of existing diffusion-based world models—their lack of long-term memory—by integrating SSM architectures to enable better temporal modeling over extended sequences.

The key contributions include: (1) a novel architecture that combines diffusion models with SSMs for world modeling, (2) a memory module trained on reward and termination signal prediction to enhance decision-making capabilities, and (3) a dynamic loss harmonization technique to balance multiple training objectives. The authors evaluate EDELINE on standard benchmarks including Atari 100k, Crafter, and selected ViZDoom tasks, demonstrating superior performance compared to baseline methods.

**Questions:**

- Line 18, "efficiencyy" looks as a typo.
- In Figure 3, the labels are too small, can you increase this?
- In line 328, it looks like there is a typo in "It architecture demonstrates ....".

**Ethical Concerns:**

["NO or VERY MINOR ethics concerns only"]

**Final Justification:**

The authors properly addressed my raised concerns, especially for the architectural innovation with their newly introduced ablation study. Hence, I maintain my positive score.

**Limitations:**

yes

**Paper Formatting Concerns:**

I did not notice any major formatting issues in this paper.

**Quality:**

4

**Strengths And Weaknesses:**

## Strengths

**Well-Motivated Research Direction.** The paper addresses a clear limitation of existing diffusion-based world models, particularly DIAMOND's inability to effectively handle long-term dependencies. Integrating memory mechanisms through SSMs represents a natural and well-justified advancement in this research area.

**Clear Problem Presentation.** Section 4 effectively demonstrates the limitations of prior work, providing concrete evidence of memory-related failures that motivate the proposed approach. This structure helps readers understand the specific problem being addressed and appreciate the necessity of the solution.

**Consistent Performance Improvements.** EDELINE demonstrates substantial and consistent performance gains over baseline methods across multiple diverse benchmarks (Atari 100k, Crafter, ViZDoom). The results provide strong empirical support for the authors' claims about the benefits of incorporating long-term memory.

**Comprehensive Experimental Analysis.** The paper provides thorough experimental validation beyond the main results, including qualitative visualizations on Atari 100k, linear probing analysis to understand learned representations, architectural ablation studies, and computational efficiency analysis. This comprehensive evaluation strengthens the paper's contributions and provides valuable insights for future work.

## Weaknesses

**Limited Architectural Innovation and Potential Redundancy.** While EDELINE achieves strong empirical results, the integration of SSMs with diffusion models appears relatively straightforward and potentially suboptimal. The architecture maintains both SSM hidden states (containing full historical context) and explicit frame stacking (last L frames) as inputs to the diffusion model, creating apparent redundancy. While this dual approach might provide benefits—such as giving additional emphasis to recent observations that are crucial for next-frame prediction—the authors provide no empirical justification for this design choice.

---

> ### Author Rebuttal · Authors · 2025-07-31
>
> We appreciate the reviewer’s valuable feedback and effort spent on the review, and would like to respond to the reviewer’s questions as follows.
>
> > Q1. Limited Architectural Innovation and Potential Redundancy.
>
> We appreciate the reviewer's observation. We would like to bring to the reviewer's attention that this represents significant architectural innovation, as evidenced by the superior performance compared to prior works in this domain. Our design choice to maintain both components stems from the complementary roles they serve in the diffusion process. The hidden embedding captures long-term temporal dependencies and global context across the entire sequence history, while the explicit frame stacking provides pixel-based spatial details from recent observations that are crucial for pixel-level prediction accuracy. The diffusion model benefits from both the abstract temporal representations (via cross-attention with hidden embedding) and direct access to recent visual features (via frame stacking) for generating high-fidelity predictions.
>
> For empirical validation to address this concern, we conducted an ablation study comparing: (1) EDELINE with both components, (2) Hidden embedding only (without frame stacking), and (3) DIAMOND. As shown in Table R4, the results demonstrate that in Atari Breakout, which requires minimal long-term memory while demanding high visual detail processing, EDELINE without frame stacking performs slightly higher than DIAMOND, though still significantly worse than full EDELINE. In MiniGrid-MemoryS9, which requires substantial memory capabilities while having simpler visual complexity, EDELINE without frame stacking can still successfully learn to solve the task and outperforms DIAMOND. Note that our analysis reveals that removing either component leads to degradation in either temporal consistency or visual fidelity.
>
> We appreciate the reviewer's suggestion and consider new architectural design or more elegant integration mechanisms as interesting avenues for future research directions. However, we would like to respectfully emphasize that our current framework achieves the primary objective of combining long-term memory with high-quality visual prediction while maintaining computational efficiency, which is the main theme of this manuscript. The experimental results faithfully validate our contributions, and we have experimental evidence demonstrating that these components are complementary in achieving superior performance.
>
> ||DIAMOND|EDELINE w/o frame-stack|EDELINE|
> |-|-|-|-|
> |MemoryS9|0.380|**0.985**|**0.982**|
> |Breakout|132.5|145.6|**250.5**|
>
> **Table R4: Frame Stacking Ablation Study.** Performance comparison demonstrating the necessity of both SSM hidden states and explicit frame stacking across environments with different memory and visual complexity requirements.
>
> > Q2. Typo in Line 18 and Line 328.
>
> We thank the reviewer for catching these typographical errors. We will correct these issues in the revised manuscript.
>
> > Q3. In Figure 3, the labels are too small.
>
> We appreciate the reviewer's feedback regarding figure readability. We will increase the label sizes in Figure 3 to enhance clarity in the revised version.

---

> > ### Comment · Reviewer_t2dk · 2025-08-05
> >
> > I would like to sincerely thank the authors for their detailed and thoughtful rebuttal. They have successfully addressed my concerns, particularly regarding the architectural innovation. The newly included ablation study on frame stacking was especially convincing. Therefore, I am happy to maintain my positive score for this paper.

---

> > > ### Author Response · Authors · 2025-08-05
> > >
> > > We sincerely thank the reviewer for the feedback. We appreciate that our rebuttal has addressed the reviewer's concerns. We deeply appreciate the time and effort the reviewer has dedicated to evaluating our work throughout this process.

---

### Official Review · Reviewer_Jqhz · 2025-06-26

**Clarity:** 3
**Significance:** 2
**Originality:** 3
**Rating:** 4
**Confidence:** 3

**Summary:**

This paper introduces EDELINE for enhancing the memory capacity of the diffusion-based world model, which is limited by the fixed context length. Specifically, EDELINE integrates the state space model with diffusion models, presenting a unified architecture of the world model. Experimental results show that EDELINE can achieve superior performance across various challenging benchmarks.

**Questions:**

### Question

1. What's the inference time of EDELINE compared to other baselines?
2. Could the authors provide additional experiments on challenging visual RL? (e.g., MineCraft, CS or dm_control)?

**Ethical Concerns:**

["NO or VERY MINOR ethics concerns only"]

**Final Justification:**

I have read the response and other reviews, and my concerns have been addressed.
This paper is well-written with convincing motivation and comprehensive experiments.
Thus, I tend to keep my positive score (Boardline Accept).

**Limitations:**

yes

**Quality:**

2

**Strengths And Weaknesses:**

### Strengths
1. The writing is clear. The research problem is well-illustrated, and the proposed method intuitively makes sense.
2. The performance of the proposed method is superior to existing SOTAs.

### Weakness
1. To further investigate the proposed world model architecture, it would be great to have the inference and training speed comparison between different world models.
2. More challenging visual RL benchmarks used in previous works can be considered (e.g., Minecraft and FPS games).

---

> ### Author Rebuttal · Authors · 2025-07-31
>
> We appreciate the reviewer’s valuable feedback and effort spent on the review, and would like to respond to the reviewer’s questions as follows.
>
> > Q1. What's the training time and inference time of EDELINE compared to other baselines?
>
> We thank the reviewer for this important question. We provide the total training time for DreamerV3 [1], IRIS [2], DIAMOND [3], and EDELINE on a single RTX 4090 GPU for Atari 100k benchmark training. This time encompasses both world model training and policy training using the world model and data collection.
>
> In addition, we provide the single-step imagination time (i.e., world model inference) for these baselines. While EDELINE's inference time is slightly higher than DIAMOND's, the adoption of Mamba's parallel scan algorithm and the random target timestep selection technique during world model training makes EDELINE more efficient in world model training compared to DIAMOND. This enables EDELINE to maintain computational efficiency comparable to DIAMOND while addressing DIAMOND's memory capacity limitations. The detailed comparison is presented in Table R2.
>
> ||DreamerV3 [1]|IRIS [2]|DIAMOND [3]|EDELINE (Ours)|
> |-|-|-|-|-|
> |Training days|**< 1**|4.1|2.9|2.9|
> |Inference Time (ms)|**13.9**|89.6|20.4|24.4|
> |Atari 100k HNS|1.124|1.046|1.459|**1.866**|
>
> **Table R2: Training and Inference Time Comparison.** Comparison of training efficiency, inference speed, and performance across different world model architectures on Atari 100k benchmark, including Atari 100k Human Normalized Score (HNS).
>
> [1] Hafner, Danijar, et al. "Mastering diverse domains through world models." Nature (2025).
>
> [2] Micheli, Vincent, Eloi Alonso, and François Fleuret. "Transformers are sample-efficient world models." ICLR (2023).
>
> [3] Alonso, Eloi, et al. "Diffusion for world modeling: Visual details matter in atari." NeurIPS (2024).
>
> > Q2. Could the authors provide additional experiments on challenging visual RL?
>
> We thank the reviewer for this suggestion. We conducted additional evaluations on the Battle scenario in ViZDoom, a 3D FPS game scenario proposed in [1]. This scenario features a 3D enclosed maze with multiple ammunition pickups, health packs, and continuously moving monsters that attack the player. The evaluation metric is the number of monsters killed by the player. We employed a sparse reward function where the agent receives a reward of 1 only upon successfully killing a monster, with the agent limited to 1M environment interactions. This setting simultaneously challenges the agent's capabilities in handling complex visual inputs, sparse rewards, 3D FPS gameplay, long-term memory, and sample efficiency. As demonstrated in Table R3 below, EDELINE significantly outperforms DIAMOND [2] due to its memory advantages and unified architecture.
>
> |Method|Avg Return|
> |-|-|
> |EDELINE (Ours)|**7.270**|
> |DIAMOND [2]|2.395|
>
> **Table R3: ViZDoom Battle Scenario Results.** Performance comparison on a challenging 3D FPS environment requiring long-term memory and sparse reward handling.
>
> [1] Dosovitskiy, Alexey, and Vladlen Koltun. "Learning to act by predicting the future." ICLR (2017).
>
> [2] Alonso, Eloi, et al. "Diffusion for world modeling: Visual details matter in atari." NeurIPS (2024).

---

> > ### Comment · Reviewer_Jqhz · 2025-08-02
> >
> > Thanks for the response, and I have read other reviews. Please consider including these additional experiments in the revised version, making the paper stronger. My concerns have been addressed properly, and I tend to keep my positive score.

---

> > > ### Author Response · Authors · 2025-08-04
> > >
> > > We sincerely thank the reviewer for the positive feedback and for taking the time to read other reviews. We are pleased to hear that our responses have adequately addressed your concerns.
> > >
> > > Thank you again for your constructive feedback throughout the review process.

---

### Official Review · Reviewer_asDt · 2025-07-02

**Clarity:** 2
**Significance:** 3
**Originality:** 2
**Rating:** 4
**Confidence:** 4

**Summary:**

This paper proposes EDELINE, a unified world model architecture that enhances long-term memory capabilities by integrating state space models (SSMs) with diffusion models. The proposed method addresses two limitations of previous diffusion-based world models like DIAMOND: (1) the inability to capture long-range temporal dependencies due to fixed-length context windows, and (2) the architectural separation of observation, reward, and termination modeling, which hinders representation sharing. EDELINE introduces three key innovations: a Recurrent Embedding Module (REM) built on the Mamba SSM for unbounded sequence modeling; a unified architecture that shares representations across all prediction heads; and a dynamic loss harmonization mechanism to balance training objectives. Experiments show that EDELINE achieves state-of-the-art performance with improved memory utilization and predictive consistency.

**Questions:**

1.	‌How does EDELINE uniquely advance beyond existing Mamba-diffusion works?

2.	Why is the input $o_{t+1}$ required for predicting $\hat{o}^0_{t+1}$ in the next-frame predictor module of Figure 2?

3.	Can the author provide the complete mathematical formulation showing how $h_t$ is derived from $h_{t-1}$ using matrices $A$, $B$, $C$, and $\Delta$ in Figure 2?

**Ethical Concerns:**

["NO or VERY MINOR ethics concerns only"]

**Final Justification:**

The authors have adequately resolved my earlier concerns regarding architectural novelty relative to prior SSM-with-diffusion works and clarified the notation. Accordingly, I raise my score to Borderline accept.

**Limitations:**

Yes.

**Paper Formatting Concerns:**

There are no formatting concerns in this paper.

**Quality:**

3

**Strengths And Weaknesses:**

Strengths:

1.	The paper conducts extensive experiments on multiple benchmarks, including Atari 100k, Crafter, and ViZDoom, demonstrating EDELINE's superior performance across diverse environments.

2.	The authors ablate key components across multiple environments and visualize predicted observations to quantify representation quality and generation fidelity.


Weaknesses:

1.	It is unclear how EDELINE differs from and improves upon existing works that also integrate SSMs with diffusion models [1] or design world models to predict multiple environment elements [2]. The authors may want to clarify the key novelty of their method.

2.	There exists symbol abuse and inconsistency in the paper. For example, the use of 'O' for both the observation space and the observation function, and the overloading of the symbol 'B' for both the matrix defined for SSMs and the burn-in period. This lack of clarity could confuse readers.

3.	In Figure 2, it is unclear why the prediction of the next frame $ \hat{o}^0_{t+1} $ requires the input $o_{t+1}$. This seems contradictory as $o_{t+1}$ is supposed to be the prediction target. The authors should clarify the data flow and the exact meanings of notations.


[1] Teng, Yao, et al. "Dim: Diffusion mamba for efficient high-resolution image synthesis." arXiv preprint arXiv:2405.14224 (2024).

[2] Lin, Jessy, et al. "Learning to model the world with language." arXiv preprint arXiv:2308.01399 (2023).

---

> ### Author Rebuttal · Authors · 2025-07-31
>
> We appreciate the reviewer’s valuable feedback and effort spent on the review, and would like to respond to the reviewer’s questions as follows.
>
> > Q1. It is unclear how EDELINE differs from and improves upon existing works that also integrate SSMs with diffusion models or design world models to predict multiple environment elements. The authors may want to clarify the key novelty of their method.
>
> We thank the reviewer for the question. Our primary contribution lies in designing a unified architecture that leverages SSMs to address the memory limitations of existing diffusion-based world models within the Model-based Reinforcement Learning domain, while maintaining computational efficiency through Mamba's parallel scan algorithm during training.
>
> We would like to clarify that most current works combining diffusion models with Mamba, such as DiM [1], DiMSUM [2], Matten [3], and other recent approaches, primarily utilize Mamba's linear time complexity and unbounded context advantages as a replacement for traditional attention-based architectures in image generation and video synthesis tasks. These works, however, concentrate primarily on architectural fusion rather than addressing the specific challenges of diffusion-based world models. Regarding world models that predict multiple environment elements like Dynalang [4], those world models focus on grounding diverse language types through GRU-based sequence modeling, while our proposed EDELINE specifically targets the memory constraints of diffusion-based world models like DIAMOND (e.g., fixed 4-frame context windows).
>
> Our key novelty lies in integrating Mamba SSMs to enable **unbounded temporal consistency** in diffusion world modeling while maintaining superior visual fidelity. This approach addresses a fundamentally different challenge from existing approaches.
>
> [1] Teng, Yao, et al. "DiM: Diffusion mamba for efficient high-resolution image synthesis." arXiv preprint arXiv:2405.14224 (2024).
>
> [2] Phung, Hao, et al. "DiMSUM: Diffusion Mamba--A Scalable and Unified Spatial-Frequency Method for Image Generation." NeurIPS (2024).
>
> [3] Gao, Yu, et al. "Matten: Video generation with mamba-attention." arXiv preprint arXiv:2405.03025 (2024).
>
> [4] Lin, Jessy, et al. "Learning to model the world with language." ICML (2024).
>
> > Q2. There exists symbol abuse and inconsistency in the paper.
>
> We appreciate the reviewer's attention to this detail. We will address these notation issues in our revised manuscript by replacing the observation space symbol $O$ with $\mathcal{O}$ and changing the burn-in length symbol $B$ to $\mathcal{B}$ to eliminate the overloading of symbols.
>
> > Q3. Why is the input $o_{t+1}$ required for predicting $\hat{o}^0_{t+1}$ in the next-frame predictor module of Figure 2?
>
> We thank the reviewer for this question. Figure 2 illustrates the **training process** of EDELINE's world model. During training, the noised version of $o_{t+1}$ ​ serves as the input to the diffusion process, where the model performs denoising based on the current conditioning information and noise level. This follows the standard training procedure for conditional diffusion models. During **inference**, no ground truth frame is available. Instead, the model generates the next frame $\hat{o}\_{t+1}$ by starting from pure Gaussian noise and progressively denoising through multiple steps without requiring $o_{t+1}$ as input. This approach is consistent with previous diffusion-based world models such as DIAMOND and follows established practices in conditional diffusion literature. We will incorporate this clarification in the revised manuscript.
>
> > Q4. Can the author provide the complete mathematical formulation showing how $h_t$ is derived from $h_{t-1}$ using matrices $A, B, C,$ and $\Delta$ in Figure 2?
>
> We thank the reviewer for this question. Our implementation follows the established Mamba literature. We concatenate the current timestep's observation $o_t$ and action $a_t$ ​as input, denoted as $u_t = (o_t, a_t)$. We project $u_t$ to initialize the input-dependent matrices $B, C$, and $\Delta$, while matrix $A$ remains input-independent. The discretization is then performed using $\Delta$ to obtain the discrete matrices $\tilde{A}$ and $\tilde{B}$. Let $x_t$ denote the Mamba latent state at timestep $t$. The Mamba architecture updates the latent state through the following recurrence:
>
> $x_t = \tilde{A} x_{t-1} + \tilde{B} u_t$
>
> The hidden embedding $h_t$ in the EDELINE architecture is obtained by applying matrix $C$ to the mamba latent state $x_t$:
>
> $h_t = C x_t$
>
> This architecture enables Mamba to store **unbounded contextual information** within the latent state and compute the current timestep's hidden embedding $h_t$ through the selective mechanism. As a result, $x_{t-1}$ contains the information from $h_{t-1}$ ​since $h_{t-1} = C x_{t-1}$, establishing the complete derivation chain from $h_{t-1}$​ to $h_t$​ through the latent state recurrence. To enhance the clarity of Figure 2, we will include the Mamba latent states $x_{t-1}$ and $x_t$​ in the revised version of the diagram.

---

> > ### Comment · Reviewer_asDt · 2025-08-02
> >
> > Thank you for the response. Regarding **Q1**, I still have some concerns: prior works in image/video synthesis such as DiM [1], DiMSUM [2], and Matten [3] also fuse Mamba-based SSMs with diffusion to leverage its efficiency and long-range modeling for high-resolution generation, which on the surface overlaps with your claim of enabling unbounded temporal consistency and superior visual fidelity.
> >
> > Could the authors clarify, from an architectural innovation perspective, how their integration differs from these prior designs, what concrete advantages those differences provide specifically for diffusion-based world modeling versus the synthesis-focused goals of the cited works, and why those advantages arise?
> >
> > [1] Teng, Yao, et al. "DiM: Diffusion mamba for efficient high-resolution image synthesis." arXiv preprint arXiv:2405.14224 (2024).
> >
> > [2] Phung, Hao, et al. "DiMSUM: Diffusion Mamba--A Scalable and Unified Spatial-Frequency Method for Image Generation." NeurIPS (2024).
> >
> > [3] Gao, Yu, et al. "Matten: Video generation with mamba-attention." arXiv preprint arXiv:2405.03025 (2024).

---

> ### Author Response · Authors · 2025-08-03
>
> We appreciate the reviewer for this follow-up question, and we would be pleased to clarify the differences.
>
> In EDELINE's architecture, Mamba serves as an independent sequence processing module (REM) that specifically handles temporal sequences of (observation, action) pairs to establish causal understanding of environment dynamics. We preserve the original diffusion model architecture while we inject Mamba's hidden embeddings as conditioning information into the diffusion process through cross-attention and adaptive group normalization (AGN). EDELINE is applied in the Model-based RL domain, with the primary goal of addressing memory limitations of previous diffusion-based world models and improving RL agent sample efficiency.
>
> We would like to bring to the reviewer's kind attention that, in contrast, works like DiM [1] primarily address computational efficiency issues of Diffusion Transformers (DiT) in high-resolution image generation. DiM tackles the fundamental challenge of adapting Mamba's 1D sequential modeling to 2D image data by introducing multi-directional scanning patterns that alternate between row-major and column-major traversals to provide global receptive fields. The method inserts learnable padding tokens at spatial boundaries to preserve image structure during sequence flattening and incorporates lightweight depth-wise convolutions for local feature enhancement. This approach enables linear computational complexity while maintaining generation quality for high-resolution image synthesis tasks. Similarly, DiMSUM [2] integrates Mamba directly into the diffusion model's core architecture with a focus on solving the spatial scanning order challenges inherent to 2D image processing. DiMSUM introduces a spatial-frequency fusion approach using wavelet transforms to capture both spatial and frequency domain information simultaneously. The architecture replaces traditional attention mechanisms within the diffusion model with Mamba-based spatial-frequency processing blocks, specifically designed to address the inductive bias limitations of Mamba when processing 2D visual data for high-quality static image generation tasks.
>
> On the other hand, Matten [3] adopts a hybrid architectural approach and directly integrates both Mamba and Attention mechanisms within the diffusion model for video generation. Matten serially combines Spatial Attention, Temporal Attention, and Global-Sequence Mamba scanning to form unified processing blocks that capture local and global spatiotemporal relationships in videos. We note that the key difference from EDELINE is that Matten embeds Mamba as a component of the diffusion model architecture directly into the generation process with a focus on spatiotemporal continuity modeling between video frames for unconditional video generation tasks.
>
> These architectural differences reflect fundamentally different design objectives. DiM and DiMSUM pursue efficient static image generation, Matten focuses on high-quality video synthesis, while EDELINE is specifically designed for causal dynamics modeling in reinforcement learning environments. Specifically, our approach requires the maintenance of memory across extended observation-action trajectories to predict future states, rewards, and terminations in partially observable environments.
>
> It is important to note that EDELINE's contribution in world modeling goes beyond visual prediction by utilizing the same Mamba hidden embeddings for multi-task learning. It simultaneously predicts next-frame observations, reward signals, and termination signals. This unified representation learning is essential for world modeling in RL, where accurate environment modeling directly impacts agent sample efficiency. In contrast, DiM and DiMSUM are specifically designed for independent image generation tasks and are not designed for world modeling scenarios that require modeling sequential state transitions and reward prediction. Similarly, Matten focuses on unconditional video generation and does not include mechanisms for modeling the causal relationships between actions and environmental outcomes necessary for world modeling.
>
> Finally, EDELINE's REM module design enables it to process unbounded sequence lengths while maintaining causal understanding of environment dynamics, which differs fundamentally from video generation tasks that primarily focus on visual coherence and quality. Therefore, EDELINE's contribution lies in specifically applying Mamba's sequence modeling capabilities to address the memory limitations of diffusion-based world models, while simultaneously enabling multi-task prediction for complete environment modeling, rather than primarily focusing on generation quality or computational efficiency.
>
> We appreciate the reviewer's thoughtful engagement and welcome further discussion to address any additional questions that may arise.

---

> > ### Author Response · Authors · 2025-08-03
> > **References**
> >
> > [1] Teng, Yao, et al. "DiM: Diffusion mamba for efficient high-resolution image synthesis." arXiv preprint arXiv:2405.14224 (2024).
> >
> > [2] Phung, Hao, et al. "DiMSUM: Diffusion Mamba--A Scalable and Unified Spatial-Frequency Method for Image Generation." NeurIPS (2024).
> >
> > [3] Gao, Yu, et al. "Matten: Video generation with mamba-attention." arXiv preprint arXiv:2405.03025 (2024).

---

> > > ### Author Response · Authors · 2025-08-07
> > >
> > > Dear Reviewer,
> > >
> > > We express our sincere gratitude for the time and effort that the reviewer has dedicated to reviewing our work. With less than three days remaining in the discussion period, we would like to respectfully inquire whether our responses have adequately addressed your questions.
> > >
> > > We would greatly appreciate the reviewer's feedback and remain available to address any additional questions that may arise.
> > >
> > >
> > > Sincerely,
> > >
> > > The Authors of Paper ID 5428

---

> > > > ### Author Response · Authors · 2025-08-09
> > > >
> > > > Dear Reviewer,
> > > >
> > > > We appreciate the reviewer's time and effort reviewing our work. With less than 24 hours left in the discussion period, we would like to kindly follow up on our rebuttal. We would greatly appreciate the reviewer's feedback and are happy to clarify any points if needed.
> > > >
> > > > We appreciate the reviewer's time and consideration.
> > > >
> > > > Sincerely,
> > > >
> > > > The Authors of Paper ID 5428

---

### Official Review · Reviewer_dLo2 · 2025-07-05

**Clarity:** 2
**Significance:** 2
**Originality:** 2
**Rating:** 4
**Confidence:** 3

**Summary:**

This paper introduces EDELINE, a unified world model architecture that integrates State Space Models (Mamba) with diffusion models to model longer action, observation sequences.  It aims to overcome the fixed context length limitation of prior methods (e.g., 4 frames for DIAMOND) and unify the training architecture end-to-end. EDELINE reports improved performance, temporal consistency, and imaginary quality across several environments.

**Questions:**

- While EDELINE theoretically supports "unbounded" sequences, the current evaluations are primarily within standard game episode lengths. What are practically sufficient context lengths needed here? Do the authors have insights on true long-horizon tasks in the real world?
- For reproducibility and to fully understand the model's scale, could the authors provide detailed specifications of the Mamba architecture used as well as the training sequence lengths utilized across the different environments?

**Ethical Concerns:**

["NO or VERY MINOR ethics concerns only"]

**Final Justification:**

I have a few main comments about this work initially:

-  I disagree with ruling out a potential candidate architecture (transformer) with a statement like this, which may misguide future explorations.
   - The authors have argued thoroughly here. Though I still would think there are no strong justification to claim Mamba saves more compute when the context is short, I also find it impractical to have the authors try all architectures when the focus of this paper is not this comparison. I think this is acceptable if the authors rephrase the statement and provide a more candid overview of the architecture choice.
- It appears the training details are not sufficiently presented. I can find some hyperparameters in the appendix, but I have a hard time finding the Mamba layer details, etc.
  - This is probably an overlook on my side while going through the appendix


In general, I still think this paper is borderline but I think an "borderline accept" is reasonable.

**Limitations:**

yes

**Quality:**

2

**Strengths And Weaknesses:**

Strengths:
 - The paper identified a context length problem in the prior work and proposes a REM module to model it.
 - The framework proposes a unified architecture which integrates reward and termination prediction within shared hidden states.

Weakness:
  - The choice of mamba, though discussed by the author, is not sufficiently supported by experiments. A few improvements would be helpful
      1. The comparison between the other REM modules supports this key design choice and should be placed in the main body instead of the appendix.
      2. It is stated that "Self-attention-based Transformer architectures, despite their strong modeling capabilities, suffer from quadratic computational complexity which impairs efficiency". This needs to be empirically justified. Transformers do suffer from quadratic complexity, but would that be a problem in this setting? For example, the examples provided discussed action step 12, which is well within Transformer's "comfort zone". And empirically, the real computation time of a Transformer might not be worse than a Mamba in many settings. Given this, this particular statement is not sufficient to rule out the choice of Transformers.
            - on a side note, if we would like to discuss how important context lengths are, I would suggest sharing stats of the context lengths used during training, and also practically useful context length in inference.
            - I am not saying Mamba is a wrong choice, but I disagree with ruling out a potential candidate architecture with a statement like this, which may misguide future explorations.
     3. It appears the training details are not sufficiently presented. I can find some hyperparameters in the appendix, but I have a hard time finding the Mamba layer details, etc.

---

> ### Author Rebuttal · Authors · 2025-07-31
>
> We appreciate the reviewer’s valuable feedback and effort spent on the review, and would like to respond to the reviewer’s questions as follows.
>
> > Q1. The comparison between the other REM modules should be placed in the main body.
>
> We thank the reviewer for this valuable suggestion. We acknowledge the reviewer's recommendation and will incorporate the comparison between different REM modules into the main body of the revised manuscript to better support our architectural design choices.
>
> > Q2. This particular statement is not sufficient to rule out the choice of Transformers.
>
> We thank the reviewer for this important observation. We agree with the reviewer that Transformer represents a viable and potentially promising architectural choice for our framework. Our initial experiments employed a window size of 16 to maintain computational feasibility and fair comparison with existing baselines during our preliminary investigations. However, we acknowledge this may have inadvertently disadvantaged Transformer architectures. When we removed this limitation and extended the window size to match Crafter's maximum episode steps (10,000), we found that Transformer performance became comparable to Mamba, as shown in Table R1.
>
> Our experimental results demonstrate that both Transformer and Mamba architectures achieve strong performance in our framework. We selected Mamba not due to its superiority over Transformer, but rather as we found that the simpler linear-time complexity solution already achieves satisfactory performance for our objectives. Given that both architectures perform well, and considering that the primary theme and most important aspect of this work is efficiency, we opted for the more computationally straightforward approach while acknowledging that Transformer remains an equally valid choice. We will update our discussion to incorporate the perspective of Transformer's capabilities in the revised manuscript.
>
> |REM Architecture|Avg Return|REM Inference FLOPs|#REM Parameters|
> |-|-|-|-|
> |Transformer|8.9 +/- 0.6|238.318M|7.251M|
> |Mamba|**11.5 +/- 0.9**|**213.307M** (-10.4%)|**5.670M** (-21.8%)|
> |Transformer w/o window limit|**10.2 +/- 0.8**|263.484M (+10.6%)|7.251M (0 %)|
>
> **Table R1: Recurrent Embedding Module (REM) Architecture Comparison on Crafter Environment.** We conducted additional comparisons between Transformer and Mamba architectures as the REM component in EDELINE. The evaluation includes performance (average return), computational efficiency (i.e., REM single-step inference FLOPs measured at context length = 32), and model complexity (i.e., number of REM parameters). The results demonstrate that both architectures are viable choices, with Transformer achieving competitive performance when the window size limitation is removed.
>
> > Q3. I would suggest sharing stats of the context lengths used during training, and also practically useful context length in inference.
>
> We thank the reviewer for the suggestion. During world model training, we employ different sequence lengths depending on the environment: 19 steps for Atari, MiniGrid, and ViZDoom, and 32 steps for Crafter. For the imagination phase when training the Actor-Critic component, we use four steps of world model burn-in followed by 15 steps of imagination horizon for Atari, MiniGrid, and ViZDoom, while Crafter employs 32 steps of burn-in with a 15-step imagination horizon. Please note that due to Mamba's characteristics, there are no specific context length limitations.
>
> > Q4. The training details are not sufficiently presented.
>
> We thank the reviewer for pointing this out. We have provided comprehensive implementation details in our supplementary materials. Algorithm 1 in Appendix E presents our overall training procedure, while Appendix D.12 contains hyperparameters used during training. In the revised version, we will include a detailed table of contents and reorganize the supplementary materials to avoid potential confusion and improve accessibility.
>
> > Q5. What are practically sufficient context lengths needed here? Do the authors have insights on true long-horizon tasks in the real world?
>
> We thank the reviewer for the question. As detailed in our response to Q3 above, the sequence lengths vary by environment and training phase. While we agree with the reviewer regarding real-world applicability opportunities, please note that the primary theme, focus, and contributions of this work concern addressing the memory limitations of existing diffusion-based world models. Our contributions enhance the potential for deploying these models in real applications, especially when long-term memory capabilities are essential for effective decision-making.
>
> > Q6. Could the authors provide detailed specifications of the Mamba architecture used as well as the training sequence lengths utilized across the different environments?
>
> We thank the reviewer for the question regarding reproducibility. We have provided Mamba-related configurations in Section D.12, including the number of Mamba layers, Mamba's latent state size, and expansion ratio. Additional settings follow standard Mamba defaults as established in the original implementation. The training sequence lengths across different environments are as described in our response to Q3 above. To further improve clarity and accessibility, we will reorganize these technical specifications for better accessibility in the revised manuscript.

---

> > ### Comment · Reviewer_dLo2 · 2025-08-03
> >
> > Thanks for providing detailed responses to all my questions.
> >
> > Regarding the mamba details: it should be my overlook for reading multiple appendices sections.
> >
> > Regarding the choice of mamba vs. Transformer: thanks to the authors for re-clarifying the choices. I  believe it is important to point out Transformer and Mamba, among other choices should be viable architecture here. I am making these suggestions from a practical perspective on whether the cost/performance tradeoff is well-considered.
> >
> > I currently still consider the paper is on the borderline.

---

> > > ### Author Response · Authors · 2025-08-04
> > >
> > > We sincerely thank the reviewer for the thoughtful engagement.
> > >
> > > First of all, we would like to bring to the kind attention of the reviewer that our primary contribution indeed addresses the practical problem of memory limitations within diffusion-based world models in the Model-based RL domain by leveraging Mamba to integrate unbounded history information into hidden embeddings. Through cross-attention and adaptive group normalization (AGN), we enable the diffusion model to effectively and efficiently utilize the rich information from these hidden embeddings for next-frame prediction. Simultaneously, we utilize the same hidden embeddings to predict reward and termination signals and naturally form a unified architecture that contrasts with DIAMOND's separated architecture approach. Moreover, our use of Harmonizers [1] to balance observation and reward prediction losses further demonstrates the advantages of this unified framework and achieves superior performance over DIAMOND across Atari, ViZDoom, and Crafter benchmarks while establishing new state-of-the-art results.
> > >
> > > From the perspective of efficiency, we note that while addressing these memory limitations, EDELINE maintains more efficient world model training than DIAMOND through Mamba's parallel scan algorithm and our proposed random target timestep selection technique. As clarified in our responses, Transformer represents a promising alternative that can achieve comparable performance to Mamba. However, it also incurs greater computational cost, as we demonstrated in Table R1 of the rebuttal where performance remains on par quadratically. We selected Mamba as the REM architecture due to its simpler linear-time complexity and satisfactory performance in our current settings. We acknowledge that exploring superior architectures represents a promising direction for future work.
> > >
> > > Through addressing the memory limitations of Diffusion World Models, EDELINE achieves substantially superior performance over DIAMOND in POMDP environments that require long-term planning and memory retention such as Crafter, and similarly demonstrates significant advantages over DIAMOND in 3D FPS environments such as ViZDoom. Our proposed solution to this memory bottleneck enables diffusion world models to compete effectively with traditional latent-space approaches such as DreamerV3 [2] while maintaining their visual advantages and opens new possibilities for applications that require both high-fidelity environment modeling and long-term memory.
> > >
> > > We believe our experimental validation across multiple challenging benchmarks, combined with our consideration of both performance and efficiency trade-offs, demonstrates the practical value of our approach for advancing diffusion-based world modeling in reinforcement learning applications.
> > >
> > > Finally, we deeply appreciate the reviewer's emphasis on practical importance, which aligns precisely with our motivation for proposing this method. We value this discussion and remain committed to engaging with any additional considerations that may arise.
> > >
> > > [1] Ma, Haoyu, et al. "Harmonydream: Task harmonization inside world models." ICML (2024).
> > >
> > > [2] Hafner, Danijar, et al. "Mastering diverse domains through world models." Nature (2025).

---

> > > > ### Comment · Reviewer_dLo2 · 2025-08-05
> > > >
> > > > I think I would agree that it is not always possible to compare all architecture choice in one paper. I highly suggest the authors to update the discussion regarding the potential of alternative architectures in the revision. While I might still consider the paper to be borderline. I may consider adjusting this to borderline accept.

---

> > > > > ### Author Response · Authors · 2025-08-07
> > > > >
> > > > > We sincerely thank the reviewer for this constructive feedback and thoughtful engagement.
> > > > >
> > > > > We will be glad to update the main manuscript with the discussion suggested by the reviewer regarding the potential of alternative architectures, specifically highlighting the potential of alternative architectures. While our choice of Mamba was driven by its linear-time complexity for computational efficiency and its effectiveness in addressing memory limitations in diffusion world models, we respectfully acknowledge that architectures such as Transformers also present valuable directions for future exploration.
> > > > >
> > > > > We deeply appreciate the reviewer's valuable feedback and engagement throughout this process, and we remain committed to addressing any additional considerations that may arise.

---

### Decision · Program_Chairs · 2025-09-17

**Decision:**

Accept (spotlight)

**Comment:**

This paper proposes a model that combines State Space Models (SSMs) with diffusion models for model-based RL.
The reviewers unanimously agreed that the paper addresses an important and well-motivated problem, a limited memory in diffusion-based world models, and that the empirical results are strong and consistent across a diverse set of challenging benchmarks.

During the discussion period, reviewers raised several questions regarding architectural choices, novelty in comparison to other diffusion works, and requests for additional computational analysis and more challenging benchmarks. The authors addressed the concerns with new experiments during the rebuttal period.

Given the importance of the problem and the comprehensive empirical evidence, I believe that this paper makes a solid contribution to the field. Thus, I recommend acceptance.